# The yeast H+-ATPase Pma1 promotes Rag/Gtr-dependent TORC1 activation in response to H+-coupled nutrient uptake

Elie Saliba[1], Minoas Evangelinos[1†], Christos Gournas[1†], Florent Corrillon[1], Isabelle Georis[2], Bruno André[1*]

[1]Molecular Physiology of the Cell, Université Libre de Bruxelles, Biopark, Gosselies, Belgium; [2]Institut de Recherches Microbiologiques J.-M. Wiame, Brussels, Belgium

**Abstract** The yeast Target of Rapamycin Complex 1 (TORC1) plays a central role in controlling growth. How amino acids and other nutrients stimulate its activity via the Rag/Gtr GTPases remains poorly understood. We here report that the signal triggering Rag/Gtr-dependent TORC1 activation upon amino-acid uptake is the coupled H+ influx catalyzed by amino-acid/H+ symporters. H+-dependent uptake of other nutrients, ionophore-mediated H+ diffusion, and inhibition of the vacuolar V-ATPase also activate TORC1. As the increase in cytosolic H+ elicited by these processes stimulates the compensating H+-export activity of the plasma membrane H+-ATPase (Pma1), we have examined whether this major ATP-consuming enzyme might be involved in TORC1 control. We find that when the endogenous Pma1 is replaced with a plant H+-ATPase, H+ influx or increase fails to activate TORC1. Our results show that H+ influx coupled to nutrient uptake stimulates TORC1 activity and that Pma1 is a key actor in this mechanism.
DOI: https://doi.org/10.7554/eLife.31981.001

*For correspondence:
Bruno.Andre@ulb.ac.be

†These authors contributed equally to this work

Competing interests: The authors declare that no competing interests exist.

## Introduction

The Target of Rapamycin Complex 1 (TORC1) plays a pivotal role in controlling cell growth in probably all eukaryotic organisms. It operates by integrating upstream signals such as growth factors (GFs) and nutrients to modulate, by phosphorylation, multiple downstream effectors, mostly proteins involved in anabolic processes (e.g. protein synthesis, ribosome biogenesis) or catabolic processes (e.g. autophagy, bulk endocytosis of plasma membrane transporters) (*González and Hall, 2017*; *Powis and De Virgilio, 2016*; *Saxton and Sabatini, 2017*). The central role of TORC1 in regulating cell growth is illustrated by the many reported cases of mTORC1 dysfunction associated with diseases including cancers (*Eltschinger and Loewith, 2016*; *Saxton and Sabatini, 2017*).

In human cells, GFs and amino acids are key signals of mTORC1 regulation. GFs act through activation of Rheb, a GTPase present at the lysosomal membrane that stimulates mTORC1 activity (*Durán and Hall, 2012*; *Zheng et al., 2014*). Amino acids such as leucine and arginine act via a complex of two GTPases, namely RagA or B and RagC or D, to promote recruitment of mTORC1 to the lysosome, where it is activated by Rheb. The heterodimeric Rag GTPase complex recruits mTORC1 to the lysosome when RagA/B is bound to GTP and RagC/D to GDP (*Kim et al., 2008*; *Sancak et al., 2008*). The activity of GTPases is typically modulated by GTPase Activating Proteins (GAPs) and Guanine nucleotide Exchange Factors (GEFs). Recent studies have aimed to identify these GAPs and GEFs and their upstream regulators and to better understand how these factors are controlled in response to variations in the cytosolic concentrations of amino acids and/or to their transport across the plasma or lysosomal membrane (*González and Hall, 2017*; *Powis and De Virgilio, 2016*; *Saxton and Sabatini, 2017*). Importantly, Sestrin and Castor proteins have recently been found to act, respectively, as cytosolic leucine and arginine sensors. When bound to their specific

**eLife digest** Cells adapt their growth rate depending on the amount of nutrients available. The protein complex called TORC1 plays a central role in this. When nutrients are abundant, TORC1 is very active and stimulates the production of proteins and other molecules needed for the cell to grow. However, when nutrients such as amino acids become scarce, TORC1 reduces its activity and allows the cells to adapt to starvation. This TORC1-mediated control of the metabolism is crucial for the cell to survive, and faulty TORC1 proteins have been associated with several diseases including cancers.

TORC1 was originally discovered in yeast, which provides a powerful model to study this control system. However, until now, it was not known how TORC1 is reactivated when amino acids are added to cells that have been starved of these molecules. Knowing the answer to this question would allow us to better understand how the availability of nutrients controls the activity of TORC1.

Now, Saliba et al. have discovered that TORC1 is not reactivated by the amino acids themselves, but by protons, which are positively charged hydrogen ions that travel into the cell together with the amino acids. This influx of protons is the driving force behind the active transport of amino acids and other nutrients into the cell, and potentially serves as a general signal to activate TORC1 in response to the uptake of nutrients, especially when cells have been starved.

Furthermore, the results showed that a specific enzyme in the cell membrane plays an essential role in activating TORC1. This enzyme pumps the protons out of the cell to compensate for their influx and to maintain the proton gradient in the membrane that drives the absorption of nutrients. When this enzyme was replaced with an equivalent plant enzyme, the proton-coupled nutrient uptake did not activate TORC1 in the yeast cells.

These findings may help scientists who are interested in how TORC1 is regulated in organisms other than mammals, such as plants or fungi. A next step will be to find out how exactly the proton pump in the cell membrane helps to activate TORC1.

DOI: https://doi.org/10.7554/eLife.31981.002

amino acids, these sensor proteins lose the ability to inhibit GATOR2, a negative modulator of the GATOR1 GAP complex which inhibits RagA/B, and this results in TORC1 activation (*Wolfson and Sabatini, 2017*). Specific lysosomal amino acid transporters and the V-ATPase complex also contribute importantly to mTORC1 control (*Goberdhan et al., 2016*; *Zoncu et al., 2011*). All this illustrates the complexity of the mechanisms through which amino acids regulate mTORC1 activity.

The TOR kinase that is part of TORC1 was originally identified in yeast, after isolation of dominant *TOR* mutations conferring resistance to rapamycin (Rap) (*Heitman et al., 1991*; *Loewith and Hall, 2011*). The protein components of TORC1, the RagA/B and C/D proteins, and their upstream GATOR-type regulatory complexes also exist in yeast (*Hatakeyama and De Virgilio, 2016*; *Loewith and Hall, 2011*). For instance, RagA/B and RagC/D correspond, respectively, to the yeast Gtr1 and Gtr2 proteins, which are part of a vacuole-associated complex (EGO) (*Dubouloz et al., 2005*) similar to the Rag-binding Ragulator of human cells (*Sancak et al., 2010*). When cells are grown in nutrient-rich medium, yeast TORC1 is active and stimulates by phosphorylation a wide variety of proteins. It notably stimulates the Sch9 kinase (*Urban et al., 2007*) under conditions promoting anabolic functions and cell growth. Active TORC1 also inhibits the Tap42-PP2A phosphatase, which stimulates autophagy, stress resistance, and nitrogen (N) transport and utilization (*Loewith and Hall, 2011*). In contrast, TORC1 is inhibited in N-starved and Rap-treated cells, so that anabolic processes, including protein synthesis, are inhibited and cell responses such as autophagy, bulk endocytosis of transporters, utilization of secondary N sources, and stress resistance are stimulated (*Hatakeyama and De Virgilio, 2016*; *Loewith and Hall, 2011*). One Tap42-PP2A target protein is the protein kinase Npr1 (Nitrogen permease reactivator 1), which is phospho-inhibited when TORC1 is active (*Schmidt et al., 1998*). Once Npr1 is inhibited, various permeases of nitrogenous compounds undergo intrinsic inactivation (*Boeckstaens et al., 2014*; *Boeckstaens et al., 2015*) or downregulation via ubiquitylation, endocytosis, and degradation (*MacGurn et al., 2011*; *Merhi and AndreAndré, 2012*).

Stimulation of TORC1 activity in yeast is usually monitored by visualizing the degree of Sch9 and/or Npr1 kinase phosphorylation. Sch9 and Npr1 are moderately phosphorylated in cells grown on a poor N source such as proline, but hyperphosphorylated upon addition of a preferential N source such as glutamine (Gln) or $NH_4^+$ (*Schmidt et al., 1998*; *Stracka et al., 2014*; *Urban et al., 2007*). In a study using Sch9 phosphorylation as readout, addition of any amino acid to proline-grown cells was found to result in rapid but transient Rag/Gtr-dependent TORC1 activation, whereas longer term TORC1 activation was observed only upon addition of an N source supporting optimal growth, for example Gln or $NH_4^+$, and it appeared not to depend on the Rag GTPases (*Stracka et al., 2014*). Furthermore, sustained activation of TORC1 in response to $NH_4^+$ is impaired in mutant cells lacking the glutamate dehydrogenases involved in assimilation of $NH_4^+$ into amino acids (*Fayyad-Kazan et al., 2016*; *Merhi and AndreAndré, 2012*). The upstream signals and molecular mechanisms involved in activation of yeast TORC1 in response to amino acid uptake and/or assimilation remain poorly known. For instance, although Gln behaves as a key signal for sustained TORC1 stimulation (*Crespo et al., 2002*; *Stracka et al., 2014*), no Gln sensor has been identified to date, and yeast seems to lack Sestrin and Castor proteins. Furthermore, no study has evidenced any particular role of vacuolar amino acid transporters in TORC1 regulation. The yeast leucyl-tRNA synthetase is reported to play a role in sensing balanced levels of isoleucine, leucine, and valine and to act as a GEF for Gtr1 (*Bonfils et al., 2012*), whereas the equivalent mammalian enzyme is proposed to control mTORC1 as a GAP for RagD (*Han et al., 2012*). On the basis of current knowledge, it would thus seem that the upstream signals and mechanisms controlling TORC1 according to the N or amino acid supply conditions might differ significantly between yeast and human cells.

The present study began with an unexpected observation regarding the uptake of β-alanine into yeast cells: this amino acid, which cannot be used as an N source (i.e. it is not a source of amino acids), stimulates TORC1 activity. Analysis of this effect has revealed that the general signal triggering Rag/Gtr-dependent activation of TORC1 in response to amino acid uptake is the influx of $H^+$ coupled to transport via $H^+$/amino-acid symporters. We further show that the Pma1 $H^+$-ATPase establishing the $H^+$ gradient at the plasma membrane is essential to this TORC1 activation, and suggest that Pma1 modulates TORC1 via signaling.

## Results

### Uptake of β-alanine via the Gap1 permease causes Rag/Gtr-dependent TORC1 activation without increasing internal pools of amino acids

In cells growing under poor N supply conditions (e.g. in a medium containing proline as sole N source), the yeast general amino acid permease Gap1 is active and stable at the plasma membrane. Under these conditions, TORC1 is only moderately active (*Schmidt et al., 1998*). Activation of TORC1 upon $NH_4^+$ uptake and assimilation into amino acids triggers Gap1 ubiquitylation, followed by its endocytosis and degradation in the vacuole (*Merhi and AndreAndré, 2012*). Thanks to isolation of a Gap1 mutant insensitive to this TORC1-dependent ubiquitylation, we have shown that substrate transport by Gap1 can also trigger Ub-dependent endocytosis and degradation of this transporter (*Ghaddar et al., 2014b*). This type of control, shared with other transporters of fungal and non-fungal species, probably enables cells to avoid excess uptake of external compounds (*Gournas et al., 2016*). To further investigate, without interference from the TORC1-dependent pathway, the mechanism of transport-elicited Gap1 ubiquitylation we sought to identify a Gap1 amino acid substrate unable to activate TORC1. We focused on beta-alanine (β-ala) and confirmed its reported status as a Gap1 substrate (*Stolz and Sauer, 1999*) by comparing the uptake of [$^{14}$C]-β-ala (0.5 mM) in proline-grown wild-type and *gap1Δ* mutant cells (*Figure 1A*). β-Ala, however, cannot sustain growth when used as sole N source, that is it cannot serve as a source of amino acids (*Figure 1A*). This contrasts with 4-aminobutyrate (GABA), an amino acid differing from β-ala by a single additional $CH_2$ group (*Figure 1A*) and whose catabolism depends on a specific GABA transaminase (*Andersen et al., 2007*). We thus tested whether β-ala transport by Gap1 triggers ubiquitylation and downregulation of the transporter. This proved to be the case, as addition of β-ala (0.5 mM) caused the appearance, above the immunodetected Gap1 signal, of two slowly migrating bands that were not observed with the non-ubiquitylable Gap1(K9R,K16R) mutant (*Figure 1B*). Upon β-ala addition, furthermore, Gap1 initially present at the cell surface underwent endocytosis

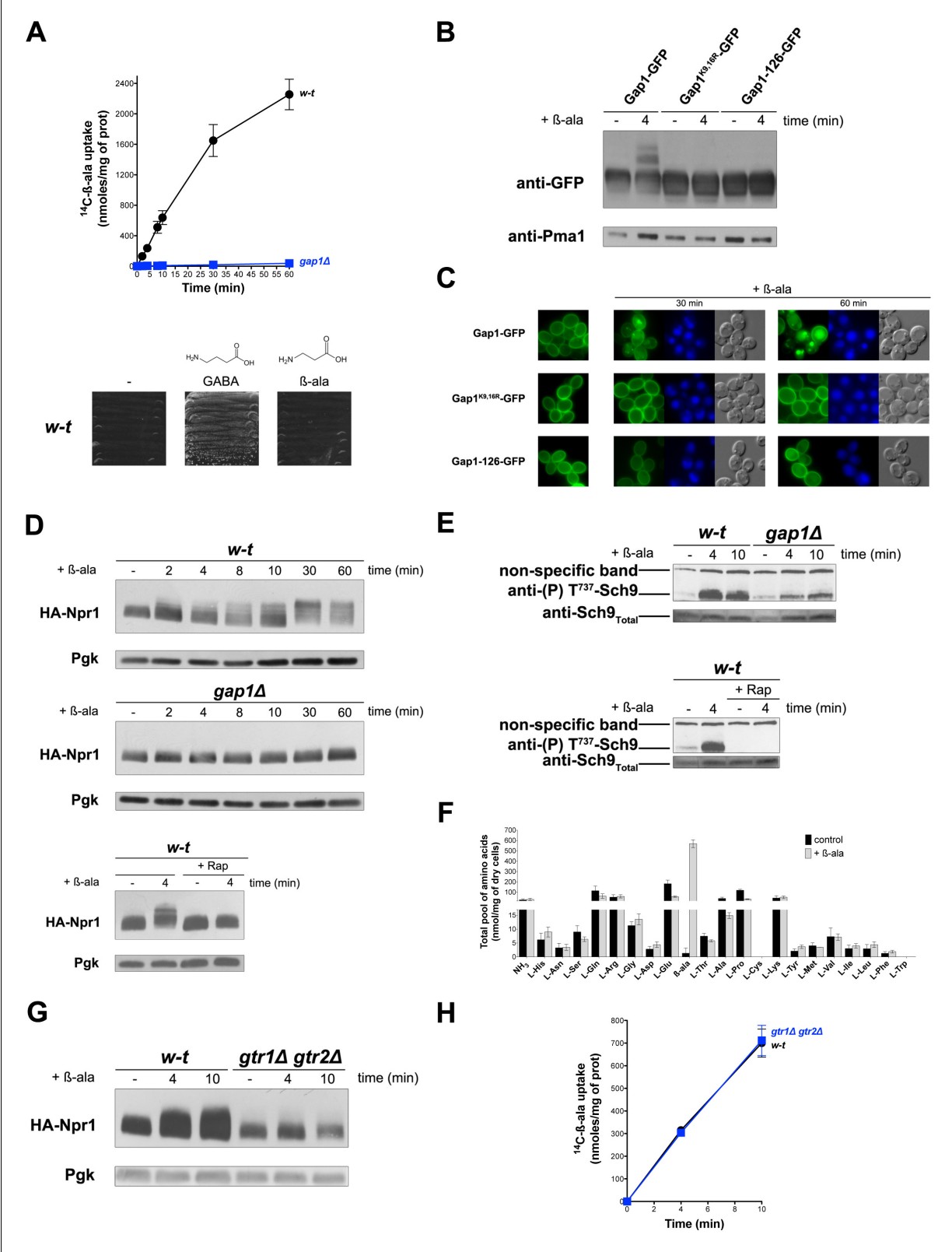

**Figure 1.** Uptake of β-alanine via the Gap1 permease causes Rag/Gtr-dependent TORC1 activation without increasing the internal pools of amino acids. (**A**) Top. Wild-type (w–t) and *gap1Δ* cells were grown on Gluc Pro medium and [$^{14}$C]-β-ala (0.5 mM) was added to the medium before measurement of the incorporated radioactivity at various times. Bottom. w-t cells were grown for 4 days on solid minimal medium without any N source or with GABA (0.5 mM) or β-ala (0.5 mM) as sole N source. (**B**) *gap1Δ* cells expressing, from plasmids, a gene encoding GFP-fused Gap1, Gap1(K9R-
*Figure 1 continued on next page*

*Figure 1 continued*

K16R), or Gap1-126 were grown on Gal Pro medium. Glucose was added for 30 min to stop Gap1 neosynthesis prior to addition of β-ala (0.5 mM) for 4 min. Crude cell extracts were prepared and immunoblotted with anti-GFP and anti-Pma1 antibodies. (C) Fluorescence microscopy analysis of the cells in 1B. Cells were grown on Gal Pro medium. Glucose was added for 1.5 hr to stop Gap1 neosynthesis, and β-ala (0.5 mM) was added for 30 min or 1 hr. CMAC staining (blue) was used to highlight the vacuole. (D) Top. w-t and *gap1Δ* cells expressing HA-Npr1 from a plasmid were grown on Gluc Pro medium. Cells were collected before and at various times after addition of β-ala (0.5 mM). Crude extracts were prepared and immunoblotted with anti-HA and anti-Pgk antibodies. Bottom. Same as in the top panel, except that w-t cells were collected before and 4 min after addition of β-ala (0.5 mM). Rap was added to half of the culture for 30 min, before addition of β-ala (0.5 mM). (E) Top. w-t and *gap1Δ* cells were grown on Gluc Pro medium. Cells were collected before and 4 and 10 min after addition of β-ala (0.5 mM). Crude extracts were prepared and immunoblotted with anti-(P) T$^{737}$-Sch9 and anti-Sch9$_{Total}$ antibodies. Bottom. Same as in panel G, except that w-t cells were collected before and 4 min after addition of β-ala (0.5 mM). Half of the culture was pretreated with Rap for 30 min. (F) w-t cells were grown on Gluc Pro medium. Cell extracts were prepared before and 30 min after addition of β-ala (0.5 mM) and used to measure amino acid pools as described under Materials and methods. The presented data are means ±SD of two independent experiments. (G) w-t and *gtr1Δ gtr2Δ* cells expressing HA-Npr1 from a plasmid were grown on Gluc Pro medium. Cells were collected before and 4 and 10 min after addition of β-ala (0.5 mM). Crude extracts were prepared and immunoblotted with anti-HA and anti-Pgk antibodies. (H) Cells as in G were grown on Gluc Pro medium. [$^{14}$C]-β-ala (0.5 mM) was added to the medium before measuring the incorporated radioactivity at various times.

DOI: https://doi.org/10.7554/eLife.31981.003

The following figure supplements are available for figure 1:

**Figure supplement 1.** Uptake of β-alanine promotes ubiquitylation and downregulation of the inactive Gap1-126 permease via TORC1-dependent stimulation of the Bul arrestins.
DOI: https://doi.org/10.7554/eLife.31981.004
**Figure supplement 2.** An anti-(P) T$^{737}$-Sch9 antibody recognizes phosphorylated Sch9 after NH$_4^+$ addition.
DOI: https://doi.org/10.7554/eLife.31981.005

and targeting to the vacuole, whereas Gap1(K9R,K16R) remained stable at the plasma membrane (*Figure 1C*). An inactive Gap1 mutant (Gap1-126) (*Ghaddar et al., 2014b*) failed to be ubiquitylated and downregulated upon β-ala addition (*Figure 1B and C*). These results are those expected if β-ala elicits Gap1 ubiquitylation specifically via the transport-elicited pathway. Yet we sought to make sure that β-ala does not activate TORC1. To our surprise, addition of β-ala to proline-grown cells caused a typical manifestation of TORC1 activation: a Rap-sensitive reduction of the electrophoretic mobility of HA-tagged Npr1, indicative of increased phosphorylation via TORC1 (*Merhi and AndreAndré, 2012*; *Schmidt et al., 1998*) (*Figure 1D*). β-Ala similarly caused a Rap-sensitive increase in the phosphorylation of Sch9 kinase residue Thr737 (*Figure 1E*), a known TORC1 target (*Urban et al., 2007*). It thus seemed that activation of TORC1, largely impaired in the *gap1Δ* mutant (*Figure 1D and E*), could contribute to the observed β-ala-induced downregulation of Gap1. This assumption was confirmed in additional experiments (*Figure 1—figure supplement 1*). Gap1-mediated uptake of β-ala thus results in TORC1 activation. We therefore hypothesized that, although β-ala cannot be used as an N source, it might be converted to certain amino acids capable of stimulating TORC1. β-Ala uptake, however, was found not to increase the intracellular concentrations of individual amino acids, measured in cell extracts, apart from that of β-ala itself (*Figure 1F*). We next tested whether this β-ala-induced activation of TORC1 involves the Rag A/B and C/D GTPases, encoded by the *GTR1* and *GTR2* genes, respectively. Increased phosphorylation of Npr1 upon β-ala addition was indeed impaired in *gtr1Δ gtr2Δ* mutant cells (*Figure 1G*), and this effect was not due to reduced uptake of β-ala (*Figure 1H*). We conclude that Gap1-mediated uptake of β-ala elicits TORC1 activation via the Rag GTPases, and that this effect is not due to conversion of intracellular β-ala to other amino acids.

## Uptake of β-ala via the endogenous Put4 or the heterologous HcGap1 permease also promotes TORC1 activation

Gap1 has been reported to be a 'transceptor', that is a protein combining the properties of transporters and receptors, capable of activating protein kinase A (PKA) in a cAMP-independent manner (*Donaton et al., 2003*). According to this model, conformational changes of Gap1, triggered by binding and/or transport of amino acids, would stimulate a PKA-targeting signaling pathway (*Schothorst et al., 2013*). We thus hypothesized that this transceptor function of Gap1 might also promote TORC1 activation in a Rag/Gtr-dependent manner. This hypothesis is potentially supported by a previous report that Gap1 interacts with Gtr2 (*Gao and Kaiser, 2006*). An alternative view is

that β-ala entering cells through Gap1 might be detected by a cytosolic amino acid sensor capable of promoting TORC1 activation. To explore these possibilities, we tested whether β-ala uptake via another permease might also activate TORC1. Confirming a previous prediction (*Gournas et al., 2015*), we found the high-affinity proline permease Put4 also to catalyze β-ala transport. This contribution of Put4 was visible at least in proline-free media, such as a medium where the sole N source was urea (another poor N source). Under these conditions, the *gap1Δ* mutant displayed residual uptake of β-ala (2 mM), and this uptake was abolished in the *gap1Δ put4Δ* mutant (*Figure 2A*). Importantly, this Put4-dependent β-ala uptake was associated with a Rap-sensitive hyperphosphorylation of Npr1 (*Figure 2B*). We next expressed in the *gap1Δ* mutant a heterologous amino acid

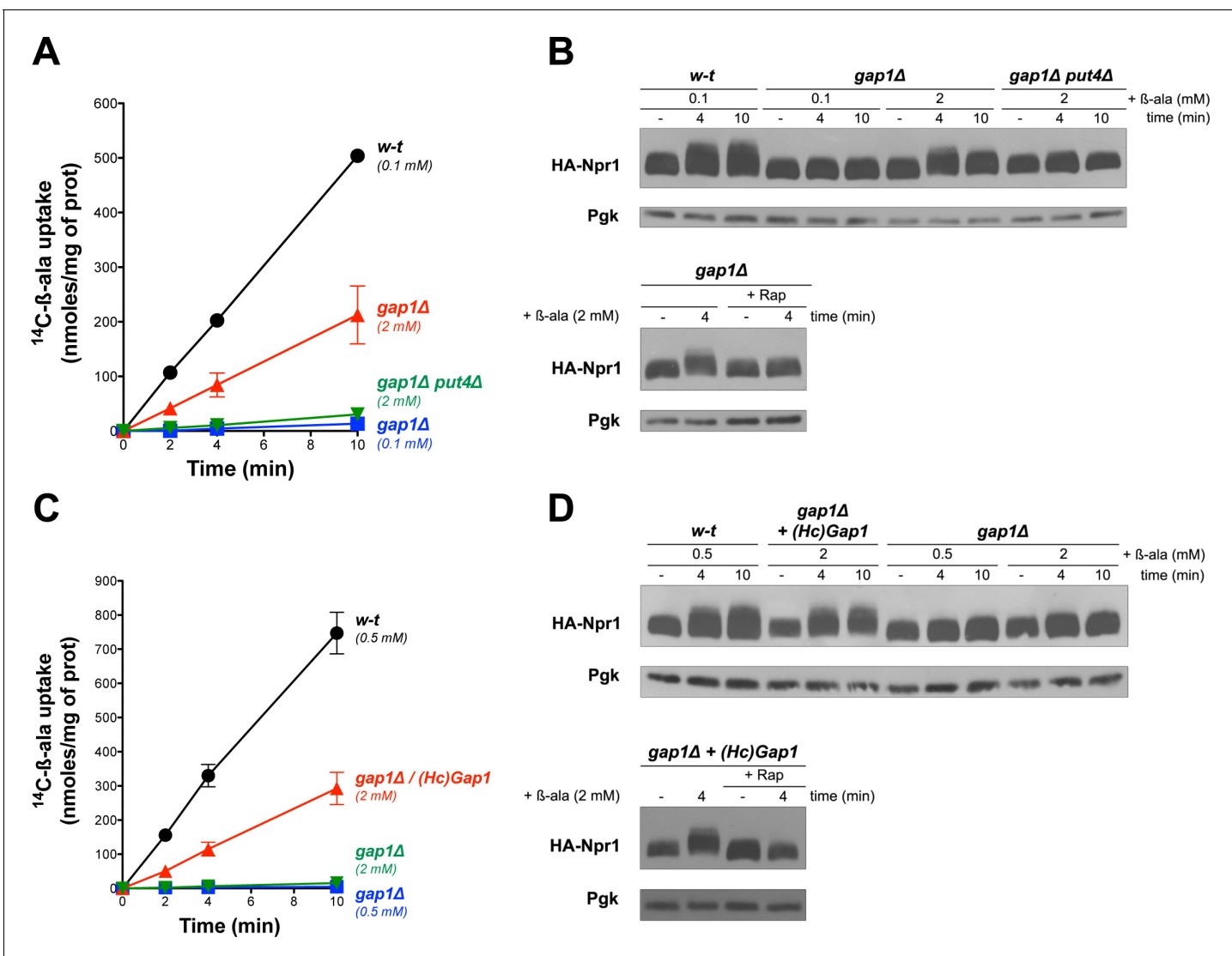

**Figure 2.** Uptake of β-ala via the endogenous Put4 or the heterologous HcGap1 permease also promotes TORC1 activation. (A) w-t, *gap1Δ*, and *gap1Δ put4Δ* cells were grown on Gluc urea medium. [$^{14}$C]-β-ala (0.1 or 2 mM) was added to the medium before measuring the incorporated radioactivity at various times. (B) Top. Cells as in A expressing HA-Npr1 from a plasmid were grown on Gluc urea medium. Cells were collected before and 4 and 10 min after addition of β-ala (0.1 or 2 mM). Crude extracts were prepared and immunoblotted with anti-HA and anti-Pgk antibodies. Bottom. Same as in panel B except that *gap1Δ* cells were also collected 30 min after Rap treatment. (C) w-t and *gap1Δ* cells expressing or not HcGap1 from a plasmid were grown on Gluc Pro medium. [$^{14}$C]-β-ala (0.5 or 2 mM) was added to the medium before measuring the incorporated radioactivity at various times. (D) Top. w-t and *gap1Δ* cells expressing HA-Npr1 and *gap1Δ* cells co-expressing HcGap1 and HA-Npr1 from plasmids were grown on Gluc Pro medium. Cells were collected before and 4 and 10 min after addition of β-ala (0.5 or 2 mM). Crude extracts were prepared and immunoblotted with anti-HA and anti-Pgk antibodies. Bottom. Same as in top panel except that cells were also treated for 30 min with Rap before β-ala addition.
DOI: https://doi.org/10.7554/eLife.31981.006

transporter known to be active in *S. cerevisiae*, namely the Gap1 permease of the fungus *Hebeloma cylindrosporum* (*Wipf et al., 2002*). HcGap1 shares ~ 30% sequence identity with Gap1 and Put4. In proline-grown *gap1Δ* cells, HcGap1 restored high β-ala uptake activity, roughly similar to that conferred by Put4 to urea-grown cells (*Figure 2C*). Remarkably, this uptake of β-ala was also associated with Rap-sensitive hyperphosphorylation of Npr1 (*Figure 2D*). In conclusion, transport of β-ala via endogenous Gap1 and/or Put4 or via the heterologous HcGap1 permease elicits TORC1 activation. Although these observations do not rule out the possibility that all three tested permeases might function as transceptors, they seem to favor the view that intracellular β-ala itself, or the process of its transport across the plasma membrane, stimulates TORC1 activity.

## Uptake of arginine via an endogenous or a heterologous permease stimulates TORC1 even if arginine catabolism is impaired

Uptake of arginine (Arg) by proline-grown cells is known to be mediated by Gap1 and the arginine-specific permease Can1 (*Wiame et al., 1985*). Consistently, we measured high Arg uptake activity in the wild type and in *gap1Δ* and *can1Δ* single mutants, but none in the *gap1Δ can1Δ* double mutant (*Figure 3A*). According to a previous report, Can1, in contrast to Gap1, does not stimulate PKA upon substrate transport (*Donaton et al., 2003*). This suggests that Can1 does not function as a transceptor. We thus sought to determine whether Arg transport via Gap1 or Can1 alone supports TORC1 activation. Arg uptake into wild-type cells was indeed found to induce Rap-sensitive Npr1 hyperphosphorylation (*Figure 3A and B*). This response was impaired in the *gtr1Δ gtr2Δ* mutant, an effect not due to decreased Arg uptake (*Figure 3B*). Arg-elicited TORC1 activation also resulted in Sch9 phosphorylation (*Figure 3C*), as previously reported (*Stracka et al., 2014*). Increased phosphorylation of Npr1 upon Arg addition was also detected in *gap1Δ* and *can1Δ* single mutants, but not in the *gap1Δ can1Δ* strain (*Figure 3A*). This shows that both permeases can promote Arg-induced TORC1 activation. We next expressed HcGap1 in the *gap1Δ can1Δ* strain and found it to restore high Arg uptake (*Figure 3D*) associated with Rap-sensitive Npr1 hyperphosphorylation (*Figure 3E*). Arginine catabolism requires arginase (Car1) (*Wiame et al., 1985*), so a *car1* mutant fails to grow on Arg as sole N source (*Figure 3F*). Arg addition to the *car1* mutant also resulted in Rap-sensitive Npr1 hyperphosphorylation (*Figure 3G*). In conclusion, TORC1 is activated upon Arg uptake via the endogenous Gap1 and/or Can1 or the heterologous HcGap1 permease. This activation of TORC1 involves the Rag GTPases and occurs even if Arg is not catabolized.

## H$^+$ influx coupled to transport promotes rag/Gtr-dependent stimulation of TORC1

The simplest way to explain the above observations is that intracellular β-ala and Arg are detected by one or several internal amino acid sensors promoting Rag/Gtr-dependent TORC1 activation. These sensors could, for instance, act like the human Castor and Sestrin proteins, recently shown to function as arginine and leucine sensors, respectively, and to modulate upstream regulators of mTORC1 (*Wolfson and Sabatini, 2017*). These sensor proteins, however, do not seem to exist in yeast. Furthermore, one would not expect a cytosolic sensor capable of activating TORC1 in response to β-ala. Alternatively, a TORC1-activating signal might arise from a common feature of the permease-mediated Arg and β-ala transport reactions. In yeast, transport by secondary active plasma membrane transporters is coupled to H$^+$ influx. This transport is thus driven by the plasma membrane H$^+$ gradient established by the Pma1 H$^+$-ATPase. We therefore hypothesized that an influx of H$^+$ coupled to amino acid uptake might initiate a signal stimulating TORC1 activity. To evaluate this hypothesis, we first checked that the β-ala and Arg transporters tested above are H$^+$-symporters. In support of this view, incubation of cells with the FCCP protonophore caused a strong reduction of β-ala uptake via Gap1, Put4, or HcGap1 and of Arg uptake via Gap1, Can1, or HcGap1 (*Figure 4A and B*). Furthermore, each permease was rapidly inhibited when the cells were shifted to glucose-free medium (*Figure 4A and B*), a condition known to cause rapid inhibition of the Pma1 H$^+$-ATPase and thus collapse of the plasma membrane H$^+$ gradient (*Kane, 2016*). Hence, as expected, all four permeases analyzed in our study, including HcGap1, behave as H$^+$-symporters. We next examined whether active uptake of another metabolite, not present in the growth medium, also elicits Rag/Gtr-dependent TORC1 activation. We chose cytosine, whose uptake via the Fcy2 permease is known to be coupled to H$^+$ influx (*Pinson et al., 1997*). Interestingly, addition of

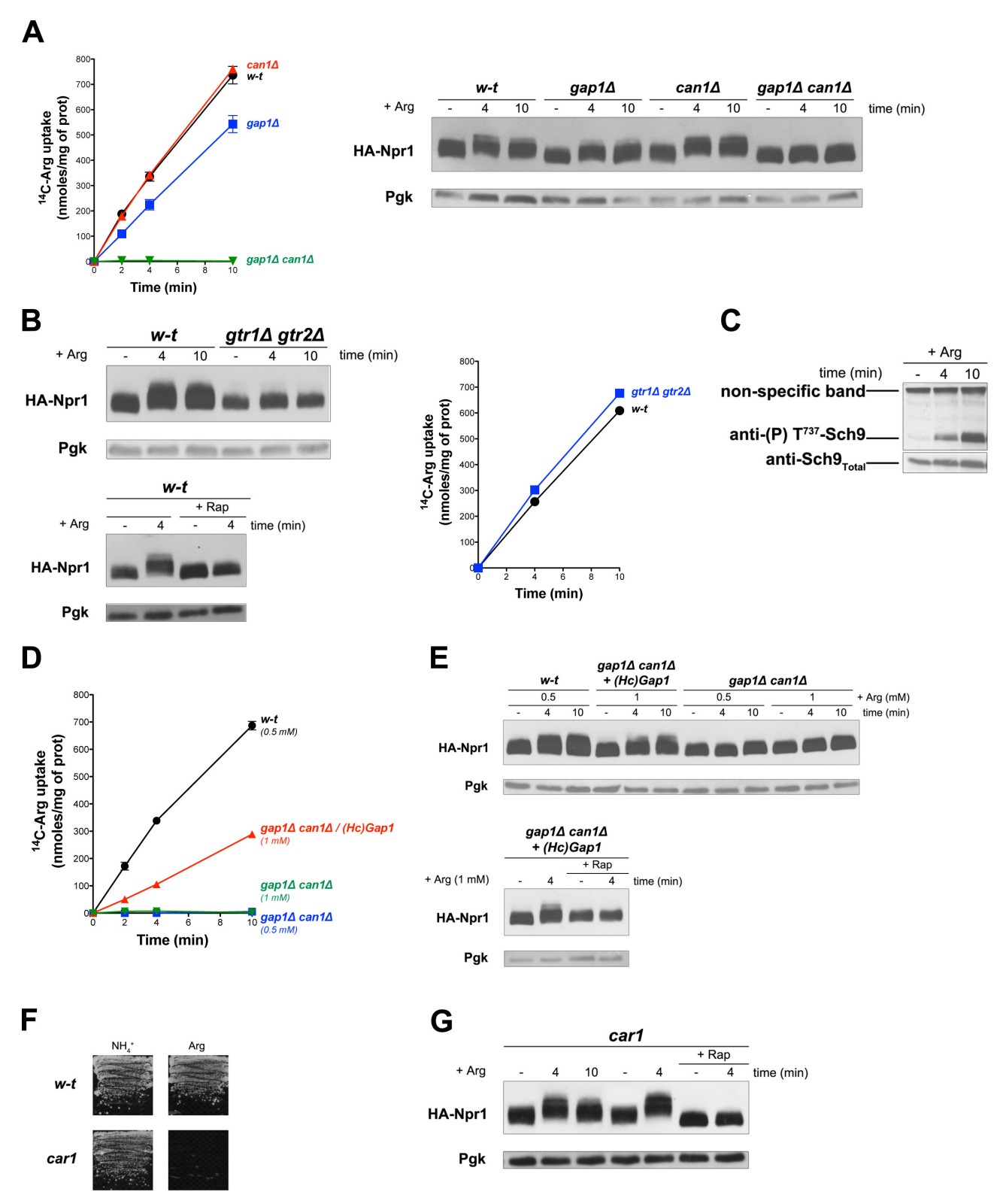

**Figure 3.** Uptake of arginine via an endogenous or a heterologous permease stimulates TORC1 even if arginine catabolism is impaired. (**A**) Left. w-t, *gap1Δ*, *can1Δ*, and *gap1Δ can1Δ* cells were grown on Gluc Pro medium. [14C]-L-Arg (0.5 mM) was added to the medium before measuring the incorporated radioactivity at various times. Right. Cells as in A expressing HA-Npr1 from a plasmid were grown on Gluc Pro medium and collected before and 4 and 10 min after addition of Arg (0.5 mM). Crude extracts were prepared and immunoblotted with anti-HA and anti-Pgk antibodies. (**B**) *Figure 3 continued on next page*

Figure 3 continued

Left top. w-t and *gtr1Δ gtr2Δ* cells expressing HA-Npr1 from a plasmid were grown on Gluc Pro medium. Cells were collected before and 4 and 10 min after addition of Arg (0.5 mM). Crude extracts were prepared and immunoblotted with anti-HA and anti-Pgk antibodies. Left bottom. Same as in panel left top except that half of the culture was treated for 30 min by Rap before addition of Arg. Right. Cells as in left panels although not expressing HA-Npr1 were grown on Gluc Pro medium. [14C]-L-Arg (0.5 mM) was added to the medium before measuring the incorporated radioactivity at various times. (C) w-t cells were grown on Gluc Pro medium. Cells were collected before and 4 and 10 min after addition of Arg (0.5 mM). Crude extracts were prepared and immunoblotted with anti-(P) $T^{737}$-Sch9 and anti-Sch9$_{Total}$ antibodies. (D) w-t and *gap1Δ can1Δ* cells expressing or not HcGap1 from a plasmid were grown on Gluc Pro medium. [14C]-L-Arg (0.5 or 1 mM) was added to the medium before measurement of the incorporated radioactivity at various times. (E) Top. w-t and *gap1Δ can1Δ* cells expressing HA-Npr1 from plasmid, and *gap1Δ can1Δ* cells co-expressing HcGap1 and HA-Npr1 from plasmids, were grown on Gluc Pro medium. Cells were collected before and 4 and 10 min after addition of Arg (0.5 or 1 mM). Crude extracts were prepared and immunoblotted with anti-HA and anti-Pgk antibodies. Bottom. Same as in top panel except that half of the culture was treated for 30 min with Rap before addition of Arg. (F) Left. w-t and *car1* mutant cells were grown on solid minimal medium with $NH_4^+$ or Arg at a concentration of 2 mM as sole N source. (G). *car1* mutant cells expressing HA-Npr1 from a plasmid were grown on Gluc Pro medium. Cells were collected before and 4 and 10 min after addition of Arg (0.5 mM), and part of the culture was also treated for 30 min with Rap before Arg addition. Crude extracts were prepared and immunoblotted with anti-HA and anti-Pgk antibodies.

DOI: https://doi.org/10.7554/eLife.31981.007

cytosine did cause rapid activation of TORC1, as judged by increased Npr1 and Sch9 phosphorylation (*Figure 4C and D*). Furthermore, this cytosine-elicited TORC1 activation was largely impaired in the *gtr1Δ gtr2Δ* mutant (*Figure 4E*). Cytosine can be used as sole N source, and thus as a source of amino acids. Yet in an *fcy1* mutant lacking cytosine deaminase and thus unable to use cytosine as an N source, Npr1 was still phosphorylated upon cytosine addition, unless Rap was also present (*Figure 4F*). $H^+$-coupled uptake of cytosine thus stimulates TORC1 in a Rag/Gtr-dependent manner, even when the nucleobase is not assimilated into amino acids. This observation is compatible with the proposed view that $H^+$ influx is the signal initiating TORC1 stimulation.

To further assess this model, we sought to analyze the activity of TORC1 upon equivalent uptake of the same external compound by either a facilitator or an $H^+$-coupled symporter. Hexoses including fructose are known to enter cells via several Hxt transporters that function as facilitators (*Wieczorke et al., 1999*). Yet particular *S. cerevisiae* strains are reported to also express an $H^+$-coupled specific fructose transporter called Fsy1 (*Galeote et al., 2010*; *Rodrigues de Sousa et al., 2004*). We thus used the *hxt* null strain, lacking the *HXT1 to −17* and *GAL2* genes and therefore unable to assimilate hexoses (*Wieczorke et al., 1999*), in which we expressed the *FSY1* gene behind its own promoter, or none hexose transporter gene, and we analyzed in parallel the wild-type (from which the *hxt* null mutant derives) expressing the endogenous Hxt facilitators. The strains were initially grown on maltose as *hxt* null cells can utilize this disaccharide. They were then shifted for a few hours to ethanol because the *FSY1* gene is more highly expressed on this carbon source (*Rodrigues de Sousa et al., 2004*). As in previous experiments, the N source was proline. Using these growth conditions, we measured equivalent 14C-fructose uptake in Hxt- and Fsy1-expressing cells (*Figure 4G*). None significant fructose uptake was detected in the *hxt* null mutant, as expected (*Figure 4G*). Furthermore, fructose uptake via Fsy1 was inhibited in the presence of FCCP, but uptake mediated by the Hxt facilitators was not (*Figure 4H*). We finally assayed TORC1 activity under these growth and fructose uptake conditions. We observed a Rap-sensitive increase of Sch9 phosphorylation upon fructose uptake by Fsy1-expressing cells. Such a TORC1 activation was observed neither in the wild-type incorporating fructose via the Hxt facilitators nor in the *hxt* null mutant expressing none fructose transporter (*Figure 4I*). TORC1 activation in response to fructose uptake thus occurred only when this transport was coupled to $H^+$ influx. This result fully supports the view that the $H^+$ influx is what generates the TORC1 activation signal.

## $H^+$ diffusion via a protonophore promotes Rag/Gtr-dependent stimulation of TORC1

In all the above experiments, TORC1 thus seems activated in response to the $H^+$-influx coupled to a nutrient transport reaction. We next determined whether the sole diffusion of $H^+$ via a protonophore such as FCCP might also elicit TORC1 activation. Using a strain stably expressing pHluorin, we first observed that the ionophore caused the cytosolic pH to drop to about 6.1, the pH of the buffered growth medium (*Figure 5A*). Remarkably, this rapid acidification of the cytosol coincided with

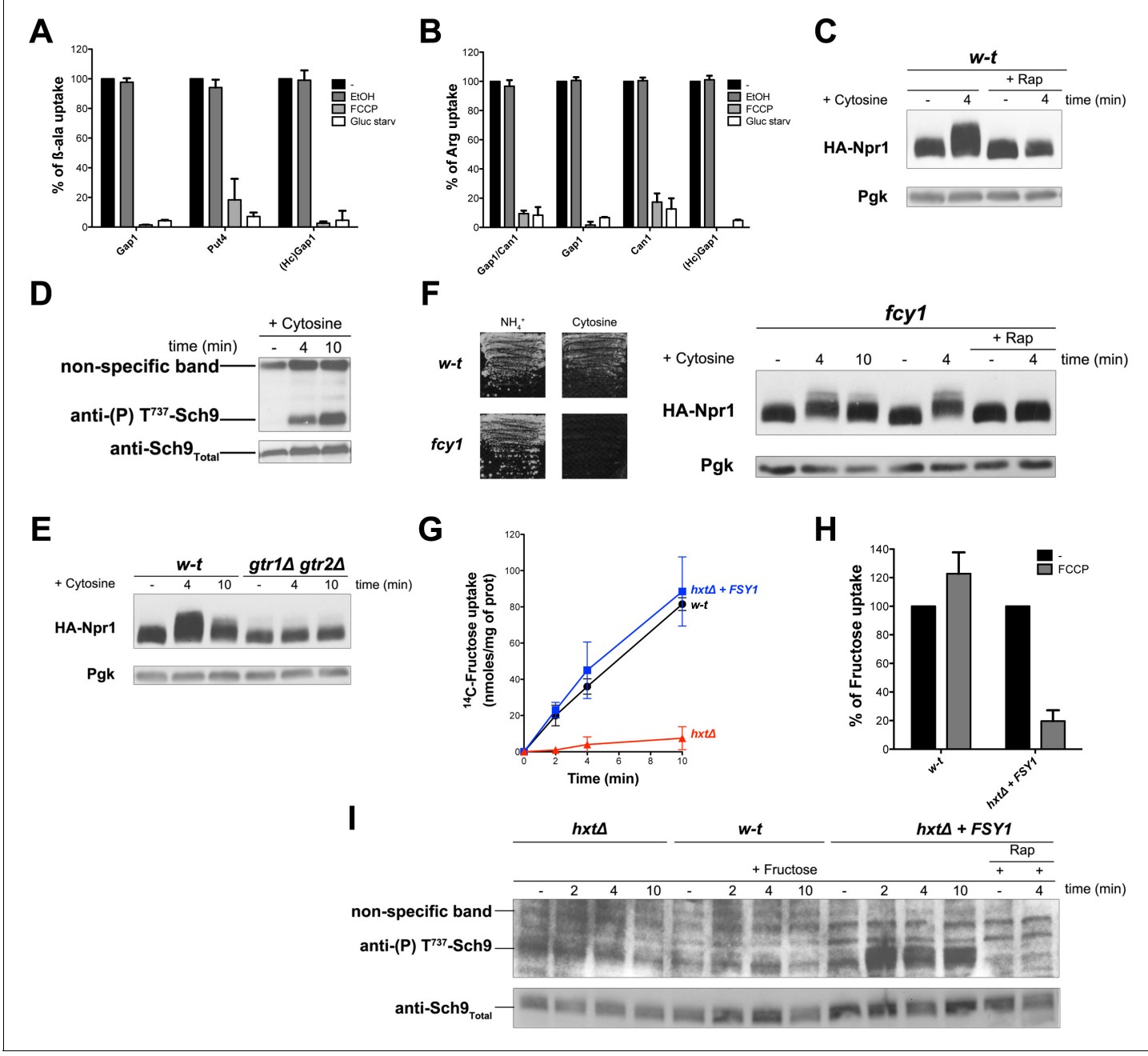

**Figure 4.** H[+]influx coupled to transport promotes Rag/Gtr-dependent stimulation of TORC1. (**A**) The β-ala uptake activities of Gap1 (in w-t cells grown on Gluc Pro medium), Put4 (in *gap1Δ* cells grown on Gluc urea medium), and HcGap1 (in *gap1Δ* cells expressing HcGap1 and grown on Gluc Pro medium), were determined by measuring the initial rate of [[14]C]-β-ala incorporation (0.5 mM for Gap1, 2 mM for Put4 and HcGap1) before and after glucose starvation for 5 min or with or without prior incubation with FCCP (20 μM) or its solvent (0.2% EtOH) for 5 min. The presented data are means ± SD of two independent experiments. (**B**) The Arg uptake activities of Gap1 and Can1 (in w-t cells), of Gap1 alone (in *can1Δ* cells), of Can1 alone (in *gap1Δ* cells), and of HcGap1 (in *gap1Δ can1Δ* cells expressing HcGap1 from a plasmid) were determined by measuring the initial rate of [[14]C]-L-Arg incorporation (0.5 mM for Gap1 and Can1, 1 mM for HcGap1) as in A. The presented data are means ± SD of two independent experiments. (**C**) w-t cells expressing HA-Npr1 from a plasmid were grown on Gluc Pro medium. Cells were collected 4 min after addition of cytosine (1 mM). Half of the culture was treated with Rap for 30 min before addition of cytosine. Crude extracts were prepared and immunoblotted with anti-HA and anti-Pgk antibodies. (**D**) w-t cells were grown on Gluc Pro medium. Cells were collected before and 4 and 10 min after addition of cytosine (1 mM). Crude extracts were prepared and immunoblotted with anti-(P) T[737]-Sch9 and anti-Sch9[Total] antibodies. (**E**) w-t and *gtr1 gtr2Δ* cells expressing HA-Npr1 from a plasmid were grown on Gluc Pro medium. Cells were collected before and 4 and 10 min after addition of cytosine (1 mM). Crude extracts were prepared and immunoblotted with anti-HA and anti-Pgk antibodies. (**F**) Left. w-t and *fcy1* cells were grown on solid medium with NH₄[+] or cytosine (2 mM) as sole N source. Right. *fcy1* cells expressing HA-Npr1 were grown on Gluc Pro medium. Cells were collected before and 4 and 10 min after

*Figure 4 continued on next page*

*Figure 4 continued*

addition of cytosine (0.5 mM). Half of the culture was treated for 30 min with Rap before addition of cytosine. Crude extracts were prepared and immunoblotted with anti-HA and anti-Pgk antibodies. (**G**) w-t (CEN.PK2-1c), *hxtΔ* (EBY.VW4000), and *hxtΔ+FSY1* (I3) cells were grown on maltose $NH_4^+$ medium, shifted for 6 hr on EtOH Pro, and [$^{14}$C]-D-fructose (2 mM) was added to the medium before measuring the incorporated radioactivity at various times. (**H**) Cells of the w-t and *hxtΔ+FSY1* strains (as in G) were grown on maltose $NH_4^+$ medium. The initial rate of [$^{14}$C]-D-fructose incorporation (2 mM), with or without prior incubation with FCCP (20 µM) for 5 min, was then measured. The data are means ±SD of two independent experiments. (**I**) Strains and growth conditions as in G. Cells were collected before and 2, 4 and 10 min after addition of fructose (2 mM). For the I3 strain, half of the culture was treated for 30 min with Rap before addition of fructose. Crude extracts were prepared and immunoblotted with anti-(P) $T^{737}$-Sch9 and anti-Sch9$_{Total}$ antibodies.

DOI: https://doi.org/10.7554/eLife.31981.008

hyperphosphorylation of Npr1, and this response was inhibited by Rap (*Figure 5B*). Hence, $H^+$ influx mediated by a protonophore also results in TORC1 stimulation. Yet this TORC1 activation, intriguingly, did not lead to increased phosphorylation of Sch9 (*Figure 5C*). A likely explanation is that an additional control elicited when the cytosol becomes too acidic (a stressful condition) impedes phosphorylation of Sch9 by activated TORC1. This is in keeping with a previous report that when the cytosolic pH drops to values around 6 (as occurs under glucose starvation or when expression of the Pma1 $H^+$-ATPase is repressed), the action of TORC1 on Sch9 is inhibited, whereas nitrogen control of the Gat1 and Gln3 transcription factors via TORC1 remains unaltered (*Dechant et al., 2014*).

TORC1 activation in response to an FCCP-mediated $H^+$ influx, thus visible on immunoblots for HA-Npr1, was largely impaired in the *gtr1Δ gtr2Δ* strain (*Figure 5D*), indicating that it is Rag/Gtr-dependent. The multisubunit SEACIT complex antagonizes TORC1 by acting as a GAP on the Gtr1 GTPase, and its function is itself negatively controlled by the multisubunit SEACAT complex (*Panchaud et al., 2013a*, *2013b*). To determine whether these GATOR-like upstream regulators of Gtr1 are involved in $H^+$ influx-elicited TORC1 activation, FCCP was added to cells lacking Seh1, a component of the SEACAT complex, or Iml1, a component of the SEACIT complexe (*Panchaud et al., 2013a*, *2013b*). TORC1 activation was largely impaired in the *seh1Δ* mutant (*Figure 5E*). In the *iml1Δ* mutant, a high basal phosphorylation of HA-Npr1 was detected, as expected, and FCCP did not significantly further increase this phosphorylation, at least during the first minutes after its addition (*Figure 5F*). These results indicate that the SEACIT/SEACAT upstream regulators of Gtr1 are involved in $H^+$-influx-elicited stimulation of TORC1 activity. We also found the amount of HA-Npr1 to be much reduced in *seh1Δ* mutant cells, and a similar effect though less pronounced was observed in the *gtr1Δ gtr2Δ* strain (*Figures 1G*, *4E* and *5D*). This suggests that the abundance of HA-Npr1 is influenced by the activation state of TORC1. Furthermore, phosphorylation of HA-Npr1 in the above-analyzed mutants was found to increase after prolonged incubation with FCCP (*Figure 5D, E and F*). It thus seems that FCCP stimulates another mechanism of TORC1 activation that does not depend on the Rag/Gtr GTPases. For instance, prolonged incubation with FCCP might promote release of amino acids from the vacuole or mitochondria, and this could promote TORC1 activation independently of Gtr1/2. It has in fact been reported that the vacuole-associated Pib2 protein containing a FYVE domain acts in parallel with Gtr1 to promote TORC1 activation (*Kim and Cunningham, 2015*; *Varlakhanova et al., 2017*). Furthermore, in an in vitro TORC1 kinase assay using isolated vacuoles, the addition of glutamine was found to stimulate TORC1 activity in a manner dependent on Pib2 but not Gtr1 (*Tanigawa and Maeda, 2017*). We thus also analyzed the role of Pib2 and found that HA-Npr1 is normally hyperphosphorylated after FCCP addition to *pib2Δ* mutant cells (*Figure 5G*).

In conclusion, the above experiments indicate that $H^+$ influx mediated even by a protonophore elicits a cellular response resulting in Rag/Gtr-dependent, Pib2-independent, TORC1 activation. They further suggest that if the cytosol becomes too acidic, an additional control likely impedes Sch9 phosphorylation by activated TORC1.

## Inhibition of the vacuolar V-ATPase activates TORC1

We observed that addition of amino acids, cytosine, or $NH_4^+$ to growing cells does not detectably change their cytosolic pH (data not shown). This was expected, given the high buffering capacity of the cytosol and the compensating $H^+$ efflux activity of the Pma1 $H^+$-ATPase, which is stimulated under acidic conditions as long as glucose is present (*Eraso and Gancedo, 1987*; *Ullah et al.,*

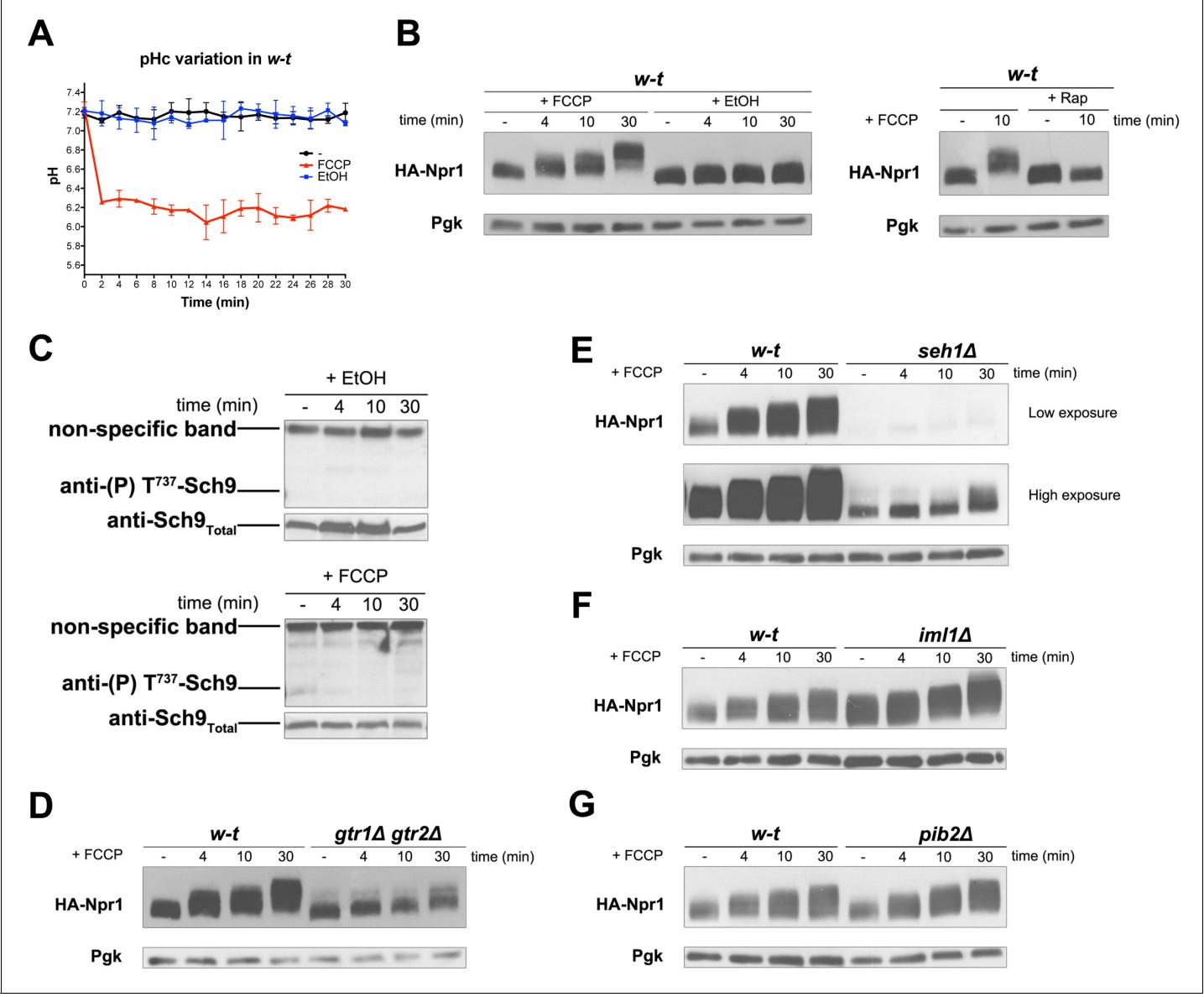

**Figure 5.** H[+]diffusion via a protonophore promotes Rag/Gtr-dependent stimulation of TORC1. (**A**) w-t cells expressing pHluorin (strain ES075) were grown on Gluc Pro medium. The cytosolic pH was monitored for 30 min at regular intervals. FCCP (20 μM) or its solvent (0.2% EtOH) was added at 1 min. (**B**) w-t cells expressing HA-Npr1 from a plasmid were grown on Gluc Pro medium. Cells were collected before and at various times after addition of FCCP (20 μM) or its solvent (0.2% EtOH). Part of the culture was treated for 30 min with Rap before addition of FCCP. Crude extracts were prepared and immunoblotted with anti-HA and anti-Pgk antibodies. (**C**) w-t cells were grown on Gluc Pro medium. Cells were collected before and at various times after addition of FCCP (20 μM) or its solvent (0.2% EtOH). Crude extracts were prepared and immunoblotted with anti-(P) T$^{737}$-Sch9 and anti-Sch9$_{Total}$ antibodies. (**D–G**) w-t and *gtr1 gtr2Δ*, *seh1Δ*, *iml1Δ* or *pib2Δ* cells expressing HA-Npr1 from a plasmid were grown on Gluc Pro medium. Cells were collected before and at various times after FCCP addition (20 μM). Crude extracts were prepared and immunoblotted with anti-HA and anti-Pgk antibodies.

DOI: https://doi.org/10.7554/eLife.31981.009

The following figure supplement is available for figure 5:

**Figure supplement 1.** Addition of the FCCP uncoupler stimulates trehalase activity in N-deprived cells.

DOI: https://doi.org/10.7554/eLife.31981.010

2012). We then examined whether an increase of cytosolic H$^+$, imposed without changing the composition of the external medium, might also lead to TORC1 activation. An H$^+$ increase can in principle be caused by inhibition of the vacuolar V-ATPase, as this enzymatic complex catalyzes ATP-dependent uptake of H$^+$ into the vacuole in order to acidify the organelle, to compensate for the constant H$^+$ efflux mediated by H$^+$-coupled vacuolar transporters, and to control the cytosolic pH (*Kane, 2016*). We thus tested the effects of two inhibitors of the V-ATPase, concanamycin A (CMA) and bafilomycin A (BAF) (*Figure 6*). Addition of CMA to proline-grown cells did not significantly change the cytosolic pH of the cells (*Figure 6A*). This suggests that Pma1-dependent efflux and the buffering capacity of the cytosol prevented the expected increase in cytosolic H$^+$. BAF addition did cause a slight but significant drop in the cytosolic pH, suggesting that this treatment caused stronger inhibition of the V-ATPase (*Figure 6E*). Remarkably, both CMA and BAF treatment resulted in stimulation of TORC1 activity, as judged by Rap-sensitive hyperphosphorylation of Npr1 (*Figure 6B* and *Figure 6F*, respectively) and by a transient increase in Sch9 phosphorylation (*Figure 6C* and *Figure 6G*, respectively). No increase in Npr1 phosphorylation was detected in either CMA- or BAF-treated *gtr1Δ gtr2Δ* mutant cells (*Figure 6D and H*, respectively). We conclude that inhibition of the V-ATPase is associated with efficient Rag/Gtr-dependent stimulation of TORC1 activity, even when this inhibition is not sufficient to cause a detectable lowering of the cytosolic pH.

## TORC1 activation in response to increased cytosolic H$^+$ requires the Pma1 H$^+$-ATPase

Activation of TORC1 in the above-described situations (H$^+$ influx, increased in cytosolic H$^+$) might involve an uncharacterized sensor of intracellular H$^+$, capable of transmitting this signal to TORC1. Alternatively, the Pma1 H$^+$-ATPase might control TORC1 activity upon sensing H$^+$ influx or increase in cytosol. For instance, Pma1 activity increases under acidic conditions, and this coincides with a reduction of its Km for ATP, possibly via allosteric control (*Eraso and Gancedo, 1987*; *Ullah et al., 2012*). We hypothesized that the particular state adopted by the H$^+$-ATPase in response to increased H$^+$ might stimulate certain factors controlling TORC1 activity. To test this possibility, we thought of expressing in yeast, instead of the endogenous Pma1, a heterologous H$^+$-ATPase known to be catalytically active in yeast. We reasoned that if TORC1 activation depends on a signaling capability of Pma1, an H$^+$-ATPase from a distant species should fail to activate TORC1 in response to an increase in cytosolic H$^+$.

According to previous reports, several plant H$^+$-ATPases are active when expressed in yeast strains where the essential *PMA1* gene and its non-essential paralog *PMA2* (expressed to a much lower level) are deleted or repressed (*Morsomme et al., 2000*; *Palmgren and Christensen, 1994*). The wild-type forms of these plant H$^+$-ATPases typically compensate only partially for the lack of Pma1. It is possible, however, to isolate mutant derivatives sustaining faster growth, particularly on low-pH media where cells need high H$^+$-ATPase activity to maintain a neutral cytosolic pH (*Morsomme et al., 2000*). For instance, the H$^+$-ATPase Pma4 of tobacco (*Nicotiana plumbaginifolia*) restores limited growth to a *pma1Δ pma2Δ* double-null mutant. A truncated Pma4 protein called Pma4$^{882ochre}$, lacking the last 71 C-terminal amino acids as a result of an ochre nonsense mutation in codon 882 of the *PMA4* gene, is able to support faster growth of yeast *pma1Δ pma2Δ* cells (*Luo et al., 1999*). We thus studied TORC1 activation in cells expressing either the endogenous *PMA1* gene or the tobacco plant *PMA4$^{822ochre}$* gene. As most the above experiments were carried out with strains having the ∑1278b background, we first isolated a *GAL1-PMA1 pma2Δ* derivative of this strain, where the *PMA1* gene is placed under the control of the galactose-inducible, glucose-repressible *GAL1* promoter. This strain can grow on galactose but not glucose, unless it contains a plasmid expressing the endogenous *PMA1* gene or the tobacco *PMA4$^{822ochre}$* gene under the control of the *PMA1* promoter (*Figure 7A*). We cultured *GAL-PMA1 pma2Δ* cells expressing *PMA1* or *PMA4$^{822ochre}$* on glucose proline medium, as in the above-described experiments. We found cells expressing *PMA4$^{822ochre}$* to grow more slowly (*Figure 7B*). This shows that the mutant plant H$^+$-ATPase does not fully compensate for the lack of Pma1. Accordingly, compared to the cytosolic pH of *PMA1*-expressing cells, that of *PMA4$^{822ochre}$*-expressing cells was slightly lower (*Figure 7C*). H$^+$-coupled uptake of β-ala (1 mM) was also significantly lower in the latter cells (data not shown). We therefore lowered the concentration of β-ala provided to *PMA1*-expressing cells in order to reach an uptake rate equivalent to that measured in *PMA4$^{822ochre}$*-expressing cells (*Figure 7D and F*). As expected, and whichever gene was expressed, [$^{14}$C]-β-ala uptake was inhibited after a brief

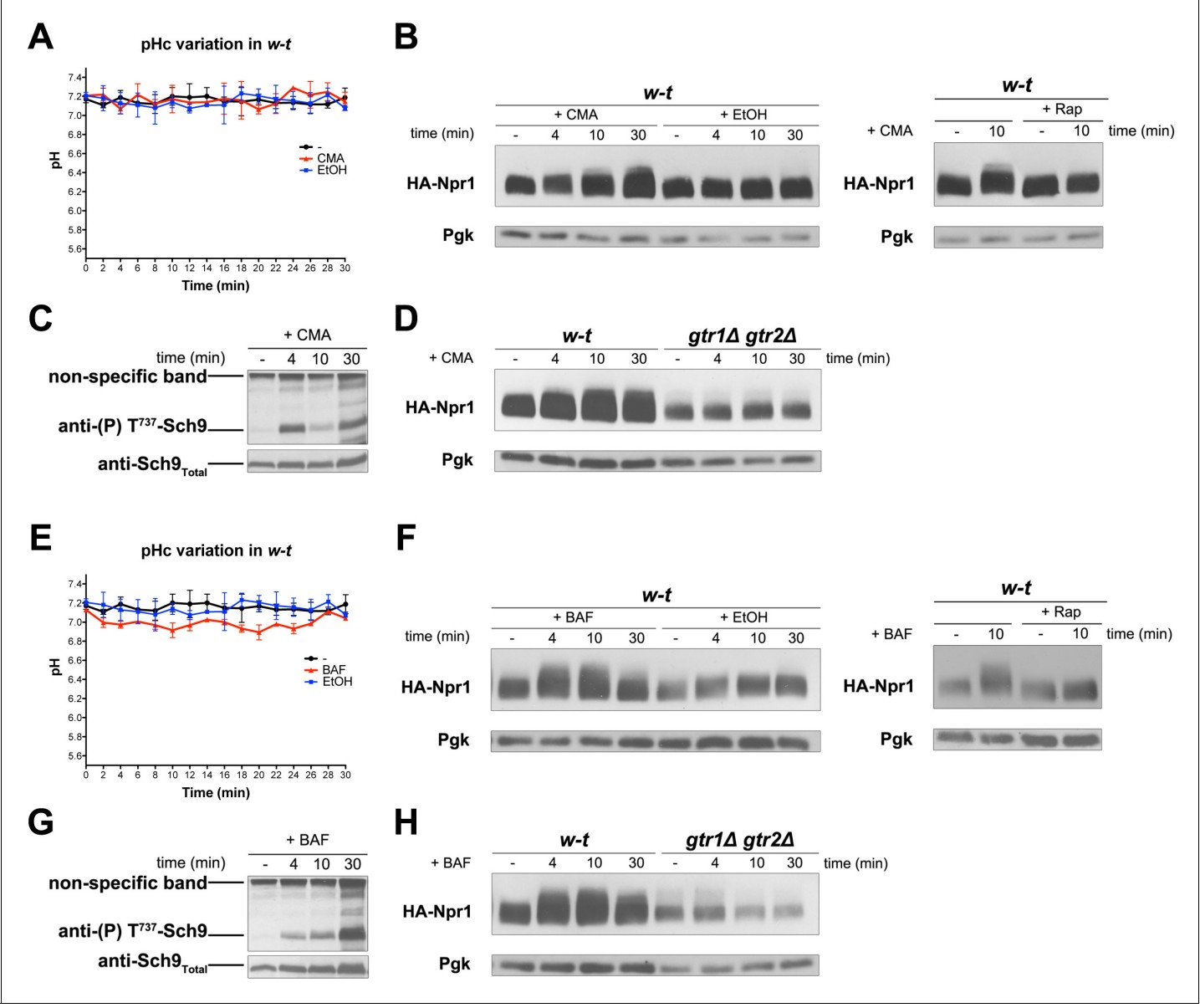

**Figure 6.** Inhibition of the vacuolar V-ATPase activates TORC1. (A) w-t cells expressing pHluorin (strain ES075) were grown on Gluc Pro medium. The cytosolic pH was monitored for 30 min at regular intervals. Concanamycin A (CMA) (1 µM) or its solvent (0.2% EtOH) was added at 1 min. (B) Left. w-t cells expressing HA-Npr1 from a plasmid were grown on Gluc Pro medium. Cells were collected before and at various times after addition of CMA (1 µM) or its solvent (0.2% EtOH). Crude extracts were prepared and immunoblotted with anti-HA and anti-Pgk antibodies. Right. Same except that half of the culture was treated with Rap for 30 min before CMA addition. (C) Same as in B except that crude extracts from w-t cells were immunoblotted with anti-(P) T$^{737}$-Sch9 and anti-Sch9$_{Total}$ antibodies. (D) Same as in B except that w-t and *gtr1Δ gtr2Δ* cells were analyzed. (E, F, G, H) Same as in A, B, C and D, respectively, except that cells were treated with bafilomycin A (BAF) (1 µM). The data of *Figure 6A and E* and those of *Figure 5A* were obtained in the same experiments but are presented in separate graphs for clarity.

DOI: https://doi.org/10.7554/eLife.31981.011

treatment of the cells with FCCP. This shows that in both cases, β-ala uptake is coupled to H$^+$ influx (*Figure 7D and F*). Upon transfer of the cells to a glucose-free medium, uptake of β-ala into *PMA1*-expressing cells was also strongly reduced (*Figure 7D and F*). This was expected, since Pma1 is inhibited under these conditions. This reduction was much less pronounced in *PMA4$^{822ochre}$*-expressing cells (*Figure 7D and F*). This result can be readily explained by the fact that inactivation of H$^+$-ATPases upon glucose starvation, a regulation conserved between yeast and plant H$^+$-ATPases, requires a C-terminal auto-inhibitory region which the truncated protein Pma4$^{822ochre}$ lacks

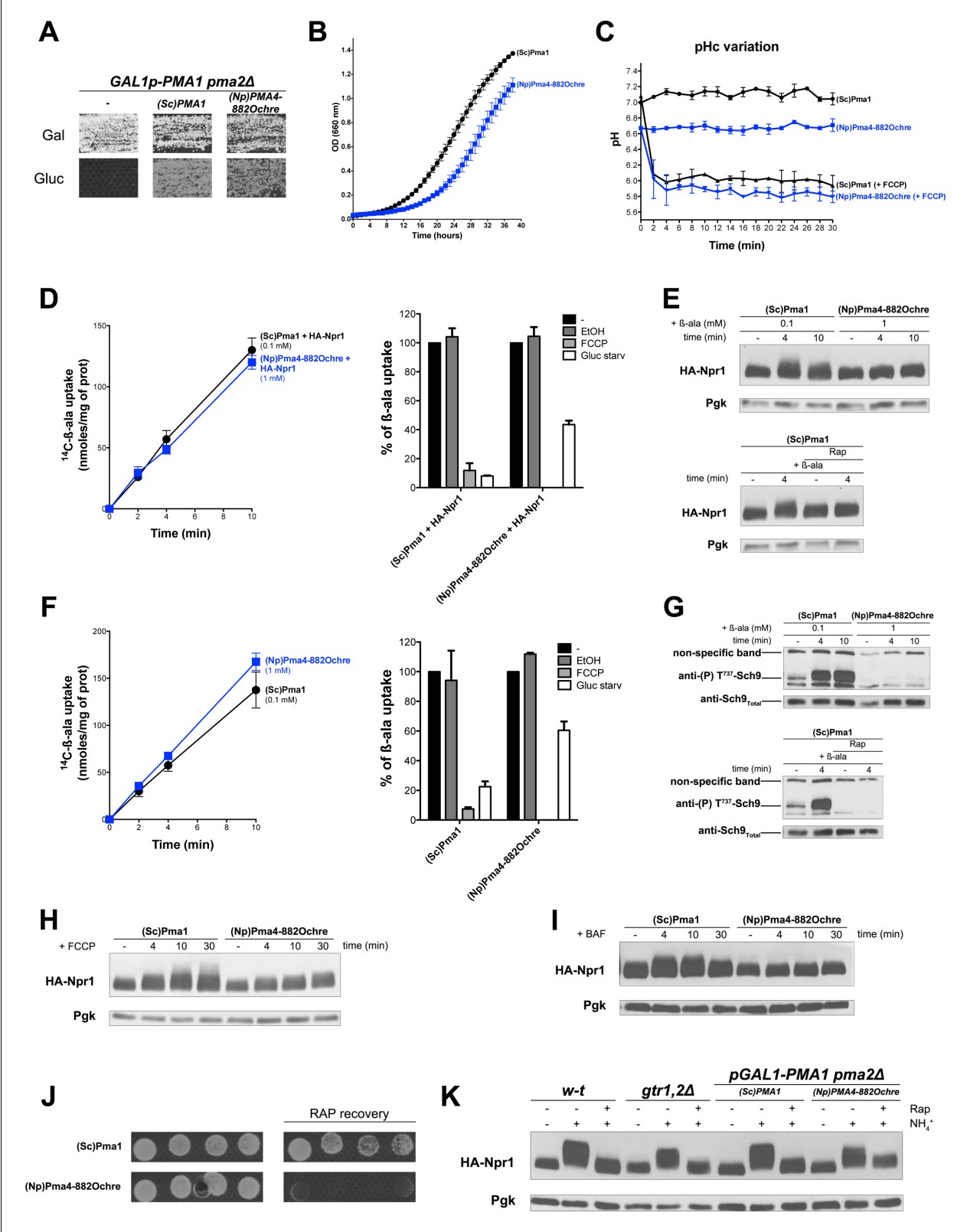

**Figure 7.** TORC1 activation in response to increased cytosolic H[+] involves the Pma1 H[+]-ATPase. (**A**) *GAL1p-PMA1 pma2Δ* cells transformed with the YCp(Sc)PMA1, YEp(Np)PMA4[882Ochre], or pFL36 (empty) plasmid (-) were grown for 3 days on solid medium with Pro as sole N source and Gal or Gluc as carbon source. (**B**) *GAL1p-PMA1 pma2Δ* cells expressing (Sc)Pma1 or (Np)Pma4[882Ochre] from a plasmid (as in A) were grown on Gluc Pro medium in a microplate reader for 40 hr. Data points represent averages of the OD at 660 nm of two biological replicates; error bars represent SD. (**C**) Strains and

*Figure 7 continued on next page*

Figure 7 continued

growth conditions as in B, except that the strains were also transformed with a plasmid (pHl-U) expressing *pHluorin*. The cytosolic pH was monitored during growth with or without addition, starting at 1 min, of FCCP (20 μM) for various times. (D) Left. Cells as in B but also expressing HA-Npr1 from a plasmid were grown on Gluc $NH_4^+$ medium. After a shift to Gluc Pro medium for 3 hr, [$^{14}$C]-β-ala (0.1 or 1 mM) was added to the medium before measuring the incorporated radioactivity at various times. Right. Strains and growth conditions as in left. Uptake via Gap1 of [$^{14}$C]-β-ala (provided at 0.1 mM to (Sc)PMA1-expressing cells and at 1 mM to (Np)PMA4$^{882Ochre}$-expressing cells) was measured before and after glucose starvation for 5 min or after addition of FCCP (20 μM) or its solvent alone (0.2% EtOH) for 5 min. Plotted values represent percentages of initial Gap1 activity, and correspond to means ± SD of two independent experiments. (E) Top. Strains and growth conditions as in D, except that the cells were collected before and 4 and 10 min after addition of β-ala (0.1 or 1 mM). Crude extracts were prepared and immunoblotted with anti-HA and anti-Pgk antibodies. Bottom. Same as in top panel, except that only (Sc)PMA1-expressing cells were collected and half of the culture was treated for 30 min with Rap. (F) Experiments similar to those in D, except that the cells harbored an empty *URA3* plasmid instead of the HA-Npr1 plasmid and were grown on Gluc Pro medium. (G) Top. Strains and growth conditions were as in F, and cells were treated as in E. Crude extracts were prepared and immunoblotted with anti-(P) T$^{737}$-Sch9 and anti-Sch9$_{Total}$ antibodies. Bottom. Same as in top panel except that (Sc)PMA1-expressing cells were collected and half of the culture was treated for 30 min with Rap. (H–I) Strains, growth conditions, and immuno-detection as in E except that cells were collected before and at various times after addition of FCCP (20 μM) or BAF (1 μM). (J) Rapamycin recovery assay for *GAL1p-PMA1 pma2Δ* cells expressing (Sc)Pma1 or (Np)Pma4$^{882Ochre}$ from plasmids. Cells were treated or not with Rap (200 ng/ml) for 6 hr, washed twice, plotted in two-fold serial dilutions on solid Gluc Pro medium, and incubated for 4 days. (K) w-t and *gtr1Δ gtr2Δ* cells expressing HA-Npr1 from a plasmid and *GAL1p-PMA1 pma2Δ* cells co-expressing (Sc)Pma1 or (Np)Pma4$^{882Ochre}$ and HA-Npr1 from plasmids were grown on Gluc $NH_4^+$ medium. After a shift to Gluc Pro medium for 3 hr, cells were collected before and 30 min after addition of $NH_4^+$ (5 mM). Cells were also collected after addition of Rap for 30 min, followed by addition of $NH_4^+$ (5 mM) for 30 min. Crude extracts were prepared and immunoblotted with anti-HA and anti-Pgk antibodies.

DOI: https://doi.org/10.7554/eLife.31981.012

The following figure supplements are available for figure 7:

**Figure supplement 1.** Pma4$^{882Ochre}$ from tobacco fails to compensate for the lack of Pma1 as regards TORC1 activation in a *pma1Δ pma2Δ* mutant
DOI: https://doi.org/10.7554/eLife.31981.013
**Figure supplement 2.** TORC1 is not stimulated by Gln, Leu, or Arg, transported into Pma4$^{882Ochre}$ expressing cells
DOI: https://doi.org/10.7554/eLife.31981.014

(*Morsomme et al., 2000*; *Portillo, 2000*). Next, under conditions of equal $H^+$-coupled β-ala uptake into *PMA1*- and *PMA4$^{822ochre}$*-expressing cells, we analyzed TORC1 activity. Phosphorylation of Npr1 was found to increase moderately upon addition of β-ala at low concentration (0.1 mM) to *PMA1*-expressing cells, but this variation was significant, as judged by its sensitivity to Rap (*Figure 7E*). Remarkably, no increase in Npr1 phosphorylation was detected in *PMA4$^{822ochre}$*-expressing cells (*Figure 7E*). Furthermore, basal phosphorylation of Npr1 in these cells did not increase, despite the lower pH of their cytosol (*Figure 7E and C*). Non-activation of TORC1 in *PMA4$^{822ochre}$*-expressing cells was even clearer when Sch9 phosphorylation was used as readout (*Figure 7G*). Basal phosphorylation of Sch9 before β-ala addition was also reduced in *PMA4$^{822ochre}$*-expressing cells (*Figure 7G*). Non-activation of TORC1 in *PMA4$^{822ochre}$*-expressing cells upon β-ala uptake was also observed in the background of another strain deleted of both *PMA1* and *PMA2* (*Figure 7—figure supplement 1*). We next analyzed TORC1 activation after addition of glutamine, leucine, or arginine, each activation being deficient in the *gtr1Δ gtr2Δ* strain (*Figure 7—figure supplement 2*). As with β-ala, the external concentration of each amino acid was first adjusted to reach equivalent uptake in *PMA1*- and *PMA4$^{822ochre}$*-expressing cells. Using these conditions, we observed a Rap-sensitive activation of TORC1 in the cells expressing *PMA1* but not in those expressing *PMA4$^{822ochre}$* (*Figure 7—figure supplement 2*). We also analyzed TORC1 activation upon FCCP addition. In both *PMA1*- and *PMA4$^{822ochre}$*-expressing cells, as expected, the ionophore caused a rapid drop in the cytosolic pH, to a value close to the pH of the buffered external medium (*Figure 7C*). Also as expected, this induced strong hyperphosphorylation of Npr1 in control *PMA1*-expressing cells (*Figure 7H*). In contrast, FCCP addition did not increase Npr1 phosphorylation in *PMA4$^{822ochre}$*-expressing cells (*Figure 7H*). Similar results were obtained after treatment with BAF: Npr1 phosphorylation was found to increase in *PMA1*-expressing but not in *PMA4$^{822ochre}$*-expressing cells (*Figure 7I*). These results show that the endogenous Pma1 $H^+$-ATPase plays an essential role in Rag/Gtr-dependent TORC1 activation in response to increased cytosolic $H^+$. The importance of Pma1 in stimulating TORC1 activity was also illustrated by the ability of *PMA1*-expressing cells to resume growth following exposure to Rap, whereas *PMA4$^{822ochre}$*-expressing cells failed to do so (*Figure 7J*).

It has been reported that the Rag GTPases are not required for sustained activation of TORC1 in the presence of $NH_4^+$ (*Stracka et al., 2014*). In support of this view, $NH_4^+$ addition to proline-grown cells caused Rap-sensitive hyperphosphorylation of Npr1 in both wild-type and *gtr1Δ gtr2Δ* mutant cells (*Figure 7K*). Furthermore, sustained TORC1 activation after $NH_4^+$ addition is reported to depend on the enzymes converting $NH_4^+$ to glutamate (*Fayyad-Kazan et al., 2016*; *Merhi and AndreAndré, 2012*), the main N donor in amino acid biogenesis reactions. Interestingly, upon $NH_4^+$ addition to proline-grown cells, we found TORC1 to be properly activated by $NH_4^+$ regardless of the $H^+$-ATPase produced (Pma1 or the plant $Pma4^{822ochre}$) (*Figure 7K*). This result shows that TORC1 can be properly activated in $PMA4^{822ochre}$-expressing cells. It also suggests that Pma1 is required for TORC1 activation in response to $H^+$ influx but not to an increase in internal amino acids.

## Discussion

Uptake of amino acids into N-deprived yeast cells causes Rag/Gtr-dependent activation of TORC1. The actual signal and the underlying mechanism of this cellular response remain unknown. Our study shows that what triggers this activation is the $H^+$ influx coupled to transport by $H^+$/amino-acid symporters. A similar response is observed upon $H^+$-dependent uptake of cytosine, or even fructose, but not when an equivalent amount of fructose enters the cells via passive transport systems. The activation of TORC1 can even be elicited by diffusion of extracellular $H^+$ via a protonophore or by inactivation of the vacuolar V-ATPase (which also causes an increase in cytosolic $H^+$). We further show that TORC1 activation in response to an $H^+$ influx and/or an increase in cytosolic $H^+$ requires the Rag GTPases encoded by the *GTR* genes. It does not, however, require the Pib2 protein recently shown to act in parallel with the Rag/Gtr proteins to activate TORC1. Finally, we show that the Pma1 $H^+$-ATPase plays a central role in this TORC1 activation pathway.

Yeast cells typically adapt to starvation for any nutrient by reducing TORC1 activity. This is probably because mechanisms capable of sensing the nutritional status of the cell impede the activation of TORC1, and not just because of a possible drop in $H^+$-coupled uptake of this nutrient. This reduction of TORC1 activity typically coincides with increased synthesis of large amounts of high-affinity $H^+$-symporters able to assimilate replenishing compounds, for example, Gap1 for amino acids, Pho84 for phosphate, Sul1 and Sul2 for sulfate, Zrt1 for zinc, Fcy2 for cytosine, and Fur4 for uracil. Furthermore, many permeases for the non-limiting nutrients likely undergo parallel increased endocytosis and degradation, as such responses have been observed in rapamycin-treated cells (*Crapeau et al., 2014*). This permease reconfiguration at the plasma membrane probably causes an overall reduction of the $H^+$ influx. Our results suggest that the $H^+$-symporters that are derepressed under starvation conditions are potentially able to reactivate TORC1 once their substrate becomes available again in the medium. In other words, $H^+$ influx via these transporters could provide a general signal for reactivating TORC1 upon relief from diverse starvation conditions. On the other hand, sustained activation of TORC1 probably requires efficient assimilation of the internalized nutrient. Accordingly, longer term TORC1 activation after addition of a preferential N source such as $NH_4^+$ is reported to require glutamine accumulation and/or synthesis (*Stracka et al., 2014*). The influx of $H^+$ coupled to uptake of any growth-limiting nutrient might thus provide a general signal for rapid, transient TORC1 reactivation in order to prepare cells for subsequent growth acceleration or restart once the nutrient has been properly assimilated. It could also contribute to the reactivation of TORC1 observed upon addition of glucose to glucose-starved cells, as although glucose enters cells via Hxt facilitators it also reactivates Pma1, which in turn drives the $H^+$-coupled uptake of amino acids and other nutrients.

Uptake of amino acids by N-starved cells is also reported to elicit transient activation of PKA, resulting in stimulation of trehalase by phosphorylation (*Donaton et al., 2003*). According to current models, PKA could be activated via a signaling pathway stimulated by Gap1 acting as a transceptor, and a similar function has been described for other $H^+$-coupled nutrient transporters found to be derepressed under particular starvation conditions (*Schothorst et al., 2013*). Yet, the mechanisms underlying this signaling remain unknown. It is tempting to envisage an alternative model according to which the signal eliciting PKA activation is the $H^+$ influx coupled to the nutrient uptake reaction, as in the case of TORC1. This model is worth considering, since addition of FCCP to N-starved cells results in stimulation of trehalase activity (*Figure 5—figure supplement 1*).

In a previous study, addition of excess Gln to proline-grown cells was found to cause a rapid, transient activation of TORC1, not observed in the *gtr1Δ* mutant, followed by a more sustained TORC1 activity, still observed in *gtr1Δ* cells (*Stracka et al., 2014*). This Gtr1-independent TORC1 activation is reminiscent of the situation described in mouse cells, where Gln activates mTORC1 in a manner independent of RagA and RagB (*Jewell et al., 2015*). Similarly, direct addition of Gln to isolated vacuoles elicits TORC1 activation in vitro in a Gtr1-independent manner (*Tanigawa and Maeda, 2017*). In contrast, this response requires the Pib2 protein proposed to act in parallel with Gtr1 to activate TORC1 (*Kim and Cunningham, 2015*; *Varlakhanova et al., 2017*). These observations suggest that Gln uptake first elicits a transient, Gtr1-dependent activation of TORC1, and we propose that the signal of this early activation is the $H^+$ influx coupled to Gln transport. The subsequent sustained activation of TORC1, in contrast, is suggested to be promoted by the intracellular accumulation of Gln (*Stracka et al., 2014*). The actual function of Pib2 in this process needs further investigation. The same applies to Gtr1/2 because, while sustained TORC1 activation occurs normally after Gln addition to *gtr1Δ* cells (*Stracka et al., 2014*), we failed to observe it in the double *gtr1Δ gtr2Δ* mutant, in keeping with another study (*Varlakhanova et al., 2017*).

We have found the Sch9 kinase to be stimulated by TORC1 in response to an increase in cytosolic $H^+$. This observation is interesting, in the light of the recent finding that Sch9 contributes to pH homeostasis by controlling the assembly and activity of the vacuolar V-ATPase (*Wilms et al., 2017*). This is relevant because the latter, together with the plasma membrane $H^+$-ATPase, contributes importantly to controlling the cytosolic pH (*Kane, 2016*). Yet according to other reports, Sch9 phosphorylation and cell growth are reduced when the cytosol becomes acidic (e.g. after a drop to a pH near 6), for instance under glucose starvation or when Pma1 synthesis is reduced (*Dechant et al., 2014*; *Orij et al., 2012*; *Ullah et al., 2012*). It thus seems that even though Sch9 is activated by an $H^+$ influx and/or by an increase in cytosolic $H^+$, it is inhibited when the cytosol becomes too acidic, and this causes growth inhibition. In FCCP-treated cells, accordingly, TORC1 appears to be efficiently activated (as judged by hyperphosphorylation of the Npr1 kinase), but this is not accompanied by Sch9 phosphorylation. This suggests that a particular mechanism sensitive to acidic conditions hampers Sch9 phosphorylation by activated TORC1. Such a control seems physiologically relevant, as acidification of the cytosol is stressful for the cell and stimulation of growth under these conditions would be inappropriate. Accordingly, other stresses are reported to promote dephosphorylation of Sch9 without affecting the TORC1-regulated Tap42-PP2A branch controlling Npr1 phosphorylation (*Hughes Hallett et al., 2014*).

A key question raised by our work is: what is the molecular mechanism responsible for stimulation of TORC1 activity in response to $H^+$ influx and/or an increase in cytosolic $H^+$? Our data indicate that the plasma membrane Pma1 $H^+$-ATPase plays a central role in this cellular response. Specifically, TORC1 activity fails to be stimulated in response to $H^+$-coupled uptake of amino acids, ionophore-mediated $H^+$ diffusion, or inhibition of the V-ATPase in cells producing the tobacco plant Pma4[822ochre] instead of Pma1. Furthermore, these cells display a strongly reduced ability to restart growth after exposure to rapamycin. At least two models can be proposed to account for these observations. On the one hand, non-activation of TORC1 might result indirectly from the inability of Pma4[822ochre] to fully compensate for the lack of Pma1 activity. We did find *PMA4[822ochre]*-expressing cells to grow more slowly and their cytosol to be slightly acidic. This might trigger adaptive feedback mechanisms impeding TORC1 reactivation. Yet in *PMA4[822ochre]*-expressing cells we found TORC1 to be properly activated after $NH_4^+$ addition, and this shows that TORC1 activity can be efficiently stimulated at least via the Rag/Gtr-independent pathway seemingly responding to internal amino acids (*Stracka et al., 2014*). Alternatively, the essential role of Pma1 in TORC1 activation in response to $H^+$ influx might reflect the ability of Pma1 to stimulate a signaling pathway controlling TORC1 activity. For instance, the activity of Pma1 is known to increase when the concentration of $H^+$ in the cytosol rises, and this control involves a decreased Km for ATP (*Eraso and Gancedo, 1987*; *Ullah et al., 2012*). This activity increase also likely occurs when protons are co-transported with nutrients via plasma membrane $H^+$-symporters. This stimulation of Pma1 activity, possibly involving a conformational change of the $H^+$-ATPase, might be transmitted to cytosolic factors that would in turn modulate TORC1 activity. A role of Pma1 in signaling to TORC1 is attractive, because Pma1 is the main ATP-consuming enzyme of yeast and is thus ideally positioned for sensing cellular ATP levels. Furthermore, as mentioned above, Pma1 stimulation by $H^+$ influx could also give cells a general mechanism for sensing relief from starvation for any nutrient and for reactivating TORC1 in response to this

relief. Other observations support a role of Pma1 in signaling to TORC1. For instance, TORC1 inhibition has been observed when Pma1 synthesis is reduced (*Dechant et al., 2014*). Furthermore, yeast TORC1 is rapidly inhibited under glucose starvation (*Urban et al., 2007*), this coinciding with polymerization of the kinase complex into a single, vacuole-associated cylindrical structure (*Prouteau et al., 2017*). Although a specific mechanism involving phosphorylation of the Kog1 subunit is reported to contribute to this TORC1 inhibition (*Hughes Hallett et al., 2015*), a role of Pma1 might also be considered, as this $H^+$-ATPase is subject to rapid and reversible auto-inhibition under these conditions (*Portillo et al., 1989*; *Serrano, 1983*). Interestingly, TORC1 has been reported recently to be required for full activity of Pma1 (*Mahmoud et al., 2017*), suggesting the existence of some crosstalk between Pma1 and TORC1. The model according to which Pma1 is capable of controlling TORC1 via signaling also seems reasonable in the light of previous works showing that the $Na^+/K^+$-ATPase of animals cells, a P-type ATPase structurally similar to Pma1 and other $H^+$-ATPases, is engaged in dynamic interactions with other proteins, including the Src tyrosine kinase. The interaction with Src is modulated by the conformation of the ion pump and initiates signal transduction processes (*Cui and Xie, 2017*). As the cytosolic region of the $Na^+/K^+$-ATPase interacting directly with Src (*Lai et al., 2013*) is relatively well conserved in the yeast $H^+$-ATPase, we introduced several substitutions into this region of Pma1 with a view of disrupting possible interactions with other factors. The generated Pma1 variants, however, behaved normally in TORC1 activation assays.

In conclusion, our results show that cytosolic $H^+$ and Pma1 are major actors in TORC1 activation in response to active nutrient uptake. They also raise the interesting possibility that Pma1 might control TORC1 via signaling. Further work is needed to evaluate this model, which would open important prospects for work on nutritional signaling in yeast and other organisms.

# Materials and methods

## Key resources table

| Reagent type (species) or resource | Designation | Source or reference | Identifiers | Additional information |
|---|---|---|---|---|
| Antibody | anti-GFP (mouse monoclonal) | Roche (Belgium) | 11814460001 ; RRID : AB_390913 | (1:10000) |
| Antibody | anti-Pma1 (rabbit polyclonal) | (*De Craene et al., 2001*) | RRID : AB_2722567 | (1:5000) |
| Antibody | anti-HA (12CA5) (mouse monoclonal) | Roche (Belgium) | 11583816001 ; RRID : AB_514506 | (1:5000) |
| Antibody | anti-Pgk (mouse monoclonal) | Invitrogen/Life technologies (Belgium) | 459250 ; RRID : AB_221541 | (1:10000) |
| Antibody | anti-(P)T737-Sch9 (rabbit purified polyclonal antibody) | This paper | RRID : AB_2722566 | GeneCust compagny; rabbit purified polyclonal antibody; against CKFAGF(pT)FVDESAIDE; (1:2500) |
| Antibody | anti-Sch9Total (rabbit polyclonal) | (*Prouteau et al., 2017*) | | (1:20000) |
| Antibody | anti-mouse IgG (whole Ab), HRP conjugate (polyclonal) | GE Healthcare/Fisher Scientific (Belgium) | NA931 ; RRID : AB_772210 | (1:10000) |
| Antibody | anti-rabbit IgG (whole Ab), HRP conjugate (polyclonal) | GE Healthcare/Fisher Scientific (Belgium) | NA934 ; RRID : AB_772206 | (1:10000) |
| Commercial assay or kit | Glucose Assay Kit | Sigma-Aldrich (Germany) | GAGO20 | Manufacture instructions |
| Chemical compound, drug | R-5000 Rapamycin | LC Laboratories (U.S.A.) | 53123-88-9 | 200 ng/ml |
| Chemical compound, drug | CellTracker Blue CMAC Dye | Life technologies (Belgium) | C2110 | |
| Chemical compound, drug | Lumi-LightPlus Western blotting substrate | Roche (Belgium) | 12015196001 | Manufacture instructions |

*Continued on next page*

*Continued*

| Reagent type (species) or resource | Designation | Source or reference | Identifiers | Additional information |
|---|---|---|---|---|
| Chemical compound, drug | Digitonin | Sigma-Aldrich (Germany) | D141 | |
| Chemical compound, drug | Carbonyl cyanide 4-(trifluoromethoxy) phenylhydrazone (FCCP) | Sigma-Aldrich (Germany) | C2920 | 20 µM |
| Chemical compound, drug | Concanamycin A (Folimycin) | Abcam (U.K.) | ab144227 | 1 µM |
| Chemical compound, drug | Bafilomycin A1 | Cell Signaling Technology (France) | 54645S | 1 µM |
| Chemical compound, drug | Alanine, β-[1–14C] | Hartmann analytic (Germany) | ARC0183 | |
| Chemical compound, drug | Arginine, L-[14C(U)] | Perkin-Elmer (Belgium) | NEC267E2 | |
| Chemical compound, drug | Leucine, L-[14C(U)] | Perkin-Elmer (Belgium) | NEC279E0 | |
| Chemical compound, drug | Glutamine, L-[14C(U)] | Perkin-Elmer (Belgium) | NEC4510 | |
| Chemical compound, drug | Fructose, D-[14C(U)] | Hartmann analytic (Germany) | ARC0116 | |
| Software, algorithm | GraphPad Prism 5 | | RRID:SCR_015807 | Statistical analysis and graphs representation |

## Yeast strains, plasmids, and growth conditions

The yeast strains used in this study (*Table 1*) derive from the Σ1278b wild type, the only exceptions being YPS14-4 (W303), JW00035 (W303), CEN.PK2-1c (VW1A), EBY.VW 4000, and I3. Cells were grown at 29°C on a minimal medium buffered at pH 6.1 (*Jacobs et al., 1980*), with glucose (Gluc) (3% w/v), maltose (3% w/v), galactose (Gal) (3% w/v), or ethanol (EtOH) (1% v/v) as a carbon source. For cultures in Gal medium, a low concentration of Gluc (0.3% w/v) was also added to boost initiation of growth. The nitrogen (N) sources added to liquid growth media were $NH_4^+$ as $(NH_4)_2SO_4$ (20 mM), proline (Pro) (10 mM), or urea (10 mM). For strain YPS14-4 and its derivative expressing $PMA4^{882-ochre}$, cells were grown on the same buffered minimal medium adjusted to pH 6.5 to improve growth. In all experiments, cells were examined or collected during exponential growth, a significant and regular number of generations after seeding. Our experience is that these precautions and the use of a minimal medium that is buffered considerably improve the reproducibility of data between biological replicates (*Wiame et al., 1985*). When indicated, rapamycin (Rap) at 200 ng/ml concentration was added for 30 min. The *ura3* mutation present in all strains was complemented by transformation with a plasmid, for example, pFL38. Comparative analyses of growth were performed by growing cells in a Greiner 24-well microplate incubator coupled to a SYNERGY multi-mode reader (BioTek Instruments). The plasmids used in this study are listed in *Table 2*.

## Fluorescence microscopy

Growing cells were laid on a thin layer of 1% agarose and viewed at room temperature with a fluorescence microscope (Eclipse E600; Nikon) equipped with a 100 differential interference contrast, numerical aperture (NA) 1.40 Plan-Apochromat objective (Nikon) and appropriate fluorescence light filter sets. Images were captured with a digital camera (DXM1200; Nikon) and ACT-1 acquisition software (Nikon) and processed with Photoshop CS (Adobe Systems). In each figure, we typically show only a few cells, representative of the whole population. Labeling of the vacuolar membrane with CMAC fluorescent dye was performed by adding 1 µl of the dye to 5 ml of culture at least 30 min prior to visualization.

**Table 1.** Yeast strains used in this study

| Strain | Genotype | Reference or source |
|---|---|---|
| 23344 c | ura3 | Laboratory collection |
| EK008 | gap1Δ ura3 | Lab collection |
| ES032 | can1Δ ura3 | (Gournas et al., 2017) |
| ES029 | gap1Δ can1Δ ura3 | (Gournas et al., 2017) |
| MA032 | gap1Δ BUL2-HA ura3 | (Merhi and AndreAndré, 2012) |
| 27038a | npi1-1$^{rsp5}$ ura3 | (Hein et al., 1995) |
| OS27-1 | bul1Δ bul2Δ ura3 | Lab collection |
| 34210 c | gap1Δ put4Δ ura3 | Lab collection |
| 33007 c | gap1Δ ura3 leu2 | Lab collection |
| 30911b | car1 ura3 | Lab collection |
| CG059 | gap1Δ can1Δ ura3 leu2 | (Gournas et al., 2017) |
| OS26-1 | gtr1Δ gtr2Δ ura3 | Lab collection |
| CEN.PK2-1c | leu2-3,112 ura3-52 trp1-289 his3-Δ1 MAL2-8c SUC2 hxt17Δ | (Wieczorke et al., 1999) |
| EBY.VW4000 | hxt1-17Δ gal2Δ stl1Δ agt1Δ mph2Δ mph3Δ leu2-3,112 ura3-52 trp1-289 his3-Δ1 MAL2-8c SUC2 | (Wieczorke et al., 1999) |
| I3 | EBY.VW4000 URA3::FSY1 | (Rodrigues de Sousa et al., 2004) |
| CG146 | seh1Δ ura3 | This study |
| CG148 | pib2Δ ura3 | This study |
| CG150 | iml1Δ ura3 | This study |
| 35652d | fcy1 ura3 | This study |
| ES075 | uga1::loxP-kanMX-loxP-GPD1p-pHluorin ura3 | This study |
| YPS14-4 | ade 2–101, leu2Δ1, his3-Δ200, ura3–52, trp1Δ63, lys2–801 pma1Δ::HIS3, pma2Δ::TRP1 + YCp-(Sc)PMA1 (LEU2) | (Supply et al., 1993) |
| PMA4–882Ochre | ade 2–101, leu2Δ1, his3-Δ200, ura3–52, trp1Δ63, lys2–801 pma1Δ::HIS3, pma2Δ::TRP1 + YEp-PMA1p-(Np)PMA4$^{882ochre}$ (LEU2) | (Luo et al., 1999) |
| JW00035 | leu2-3-112 ura3-1 trp1-1 his3-11-15 ade2-1 can1-100 sch9Δ::TRP1 | (Wilms et al., 2017) |
| JX023 | GAL1p-PMA1 pma2Δ ura3 leu2 | This study |

DOI: https://doi.org/10.7554/eLife.31981.015

## Protein extracts and western blotting

For western blot analysis, crude cell extracts were prepared as previously described (Hein et al., 1995). Proteins were transferred to a nitrocellulose membrane (Schleicher and Schuell; catalog number NBA085B) and probed with mouse anti-GFP (Roche; catalog number 11 814 460 001), anti-hem-agglutinin (anti-HA) (12CA5; Roche), or anti-yeast 3-phosphoglycerate kinase (anti-PGK) (Invitrogen) or with rabbit anti-Pma1 (De Craene et al., 2001), anti-phospho-Thr$^{737}$-Sch9, or anti-Sch9$_{Total}$ (see below). Primary antibodies were detected with horseradish-peroxidase-conjugated anti-mouse or anti-rabbit immunoglobulin G secondary antibodies (GE Healthcare), followed by enhanced chemilu-minescence (Roche; catalog number 12 015 196 001). Each Western blot was carried two to four times, a representative experiment is presented.

## Generation and validation of the anti-phospho-Thr737-Sch9 antibody

The antibody was produced by and purchased from the GeneCust company. The CKFAGF(pT)FVDE-SAID peptide containing phosphorylated Thr737 was injected into rabbit. The affinity of the anti-body preparation was tested in an ELISA for the phosphorylated peptide. Antibody specificity was tested by western blot analysis of cell extracts isolated from proline-grown wild-type (w-t) and sch9Δ strains, before and after addition of NH$_4^+$, well known to stimulate Sch9 phosphorylation (*Figure 1—figure supplement 2*). The anti-Sch9$_{Total}$ antibody was a kind gift of Robbie Loewith.

**Table 2.** Plasmids used in this study

| Plasmid | Description | Reference or source |
|---|---|---|
| pFL38 | CEN-ARS (URA3) | (*Bonneaud et al., 1991*) |
| pFL36 | CEN-ARS (LEU2) | (*Bonneaud et al., 1991*) |
| p416 GAL1 | CEN-ARS GAL1p (URA3) | (*Mumberg et al., 1994*) |
| pJOD10 | p416 GAL1p-GAP1-GFP (URA3) | (*Nikko et al., 2003*) |
| pCJ038 | p416 GAL1p -GAP1(K9R,K16R)-GFP (URA3) | (*Lauwers and André, 2006*) |
| pMA065 | p416 GAL1p -GAP1-126-GFP (URA3) | (*Merhi et al., 2011*) |
| pMA091 | p416 GAL1p -GAP1-126-K9R,K16R-GFP (URA3) | (*Merhi et al., 2011*) |
| pCH500 | CEN-ARS-GAP1 (URA3) | (*Hein and André, 1997*) |
| pHcGAP1 | YEp-HcGAP1 (URA3) | (*Wipf et al., 2002*) |
| pES103 | YEp-HA-NPR1 (LEU2) | This study |
| pAS103 | YEp-HA-NPR1 (URA3) | (*Schmidt et al., 1998*) |
| pCJ315 | CEN-ARS (LEU2-HIS3-LYS2) | Lab collection |
| pMYC008 | YCp-AGP1p-LACZ (HIS3 TRP1 LEU2) | Lab collection |
| pES154 | YCp-AGP1p-LACZ (HIS3 TRP1) | This study |
| pHl-U | YEp-ADH1p-pHluorin (URA3) | (*Orij et al., 2009*; *Zimmermannova et al., 2015*) |
| pHl-I | YCp-loxP-kanMX-loxP-GPD1p-pHluorin (URA3) | (*Orij et al., 2009*; *Zimmermannova et al., 2015*) |
| pCJ366 | YEp (TRP1-LEU2-HIS3) | Lab collection |
| pPS15-P1 | YCp-(Sc)PMA1 (LEU2) | (*Supply et al., 1993*) |
| pPMA4882ochre | YEp-PMA1p-(Np)PMA4$^{882ochre}$ (LEU2) | (*Luo et al., 1999*) |

DOI: https://doi.org/10.7554/eLife.31981.016

## Measurements of cytosolic pH

Yeast strains expressing a single *pHluorin* gene integrated into the genome or containing a multi-copy plasmid expressing the *pHluorin* gene were grown at 29°C on Gluc proline buffered medium, pH 6.1, to OD660 ~0.2. Fluorescence intensities were recorded with a SYNERGY multi-mode micro-plate reader (BioTek Instruments) with emission filter 512/9 nm and excitation filters 395/9 and 475/9 nm, as previously reported (*Orij et al., 2009*; *Zimmermannova et al., 2015*). To eliminate the background fluorescence, pHluorin-nonproducing wild-type cells were grown in parallel, and the corresponding values at each excitation wavelength were subtracted from those of pHluorin-producing cells. The I395 nm to I475 nm emission intensity ratio was used to calculate the cytosolic pH. The fluorescence intensities of each strain were typically recorded in four separate cultures (1 ml culture per well) within one experiment (technical replicates), and the presented data are means ±SD of at least two independent experiments (biological replicates). The calibration curve was generated as described previously (*Orij et al., 2009*; *Zimmermannova et al., 2015*), with minor changes. The cell culture (100 ml, OD660 = 0.2) was filtered, washed, resuspended in 8 ml phosphate-buffered saline (Sigma) containing digitonin (175 µg/ml), and incubated for 15 min at RT. Digitonin was washed out and the cells were resuspended in 8 ml PBS (the OD of the cell suspension was about 2.5) and placed on ice. Then 40 µl aliquots were transferred to CELLSTAR black polystyrene clear-bottom 96-well microtiter plates (Greiner Bio-One) containing, per well, 160 µl citric acid/Na2HPO4 buffer at a pH ranging from 5.6 to 7.6 (in this volume, the OD was 0.5). Recording of pHluorin fluorescence emission and background subtraction were performed as described above. The I395 nm to I475 nm intensity ratio was calculated, plotted against the corresponding buffer pH, and fitted to a third-degree polynomial regression curve.

## Uptake measurements of radiolabelled compounds

The accumulation of [$^{14}$C]-labeled amino acids or [$^{14}$C]-labeled-fructose was measured at the time points indicated as previously described (*Ghaddar et al., 2014a*; *Grenson et al., 1966*). The

radiolabeled compounds were purchased either from Perkin-Elmer or from Hartmann analytic. Data points represent averages of two biological replicates; error bars represent standard deviations (SD).

## Measurement of total amino acid pools

Yeast cultures (50 ml) were collected by centrifugation (7000 g for 3 min) and washed twice with 10 ml Milli-Q water. The final pellet was resuspended in 2 ml Milli-Q water and boiled for 15 min. To remove cell debris, suspensions were centrifuged at 13,000 g for 1 min and filtered (Millipore 0.45 µm). The resulting soluble fractions were subjected to amino acid analysis after AccQ Tag pre-column derivatization (Waters). For this an AccQ Tag Ultra UPLC column (Waters) with UV detection at 260 nm was used according to the manufacturer's recommendations (*Fayyad-Kazan et al., 2016*).

## Assay of trehalase activity in N-deprived cells

Cells growing on Gluc $NH_4^+$ medium were collected by filtration, and after washing and resuspension in Gluc medium without any N source, they were incubated overnight at 29°C with shaking. Cells were filtered, washed, and transferred again to fresh N-free Gluc medium for 30 min before addition of FCCP (20 µM). Culture samples were collected at various times and trehalase activity was measured in permeabilized cells as previously described (*De Virgilio et al., 1991*). Glucose levels were measured using the 'Glucose assay kit' (Sigma-Aldrich, Belgium).

## Acknowledgements

We are grateful to Pierre Morsomme, Marc Boutry, Olga Zimmermannova, Hana Sychrova, Paula Gonçalves, Joris Winderickx, Wolf Frommer, Daniel Wipf, and Eckhard Boles for strains and plasmids, and Robbie Loewith for the antibody against Sch9. We also thank Catherine Jauniaux, Charlotte Felten, and Elizabeth Bodo for skillful technical assistance, Pierre Morsomme, Marc Boutry, and all lab members for fruitful discussions, and Eckhard Boles for advices about the experiments using the *hxt* null strain. E Saliba is a fellow of the Fonds pour la Formation à la Recherche dans l'Industrie et l'Agriculture (FRIA). CGournas and M Evangelinos are postdoctoral researchers of the Fonds de la Recherche Scientifique (FRS-FNRS). This work was supported by an FNRS grant (3.4.592.08 .F) and the Algotech program of the Fédération Wallonie Bruxelles.

## Additional information

### Funding

| Funder | Grant reference number | Author |
|---|---|---|
| Fonds National de La Recherche Scientifique | 3.4.592.08.F | Bruno André |
| Fonds pour la Formation à la Recherche dans l'Industrie et dans l'Agriculture | 21074048 | Elie Saliba |
| Fonds National de La Recherche Scientifique | 22396499 | Christos Gournas |
| Fonds National de La Recherche Scientifique | 30274494 | Minoas Evangelinos |

The funders had no role in study design, data collection and interpretation, or the decision to submit the work for publication.

### Author contributions

Elie Saliba, Conceptualization, Validation, Investigation, Methodology, Writing—original draft, Writing—review and editing; Minoas Evangelinos, Christos Gournas, Florent Corrillon, Conceptualization, Validation, Investigation; Isabelle Georis, Conceptualization, Validation, Investigation, Methodology; Bruno André, Conceptualization, Supervision, Funding acquisition, Validation, Writing—original draft, Project administration, Writing—review and editing

**Author ORCIDs**

Bruno André http://orcid.org/0000-0001-7683-9150

**Decision letter and Author response**

Decision letter https://doi.org/10.7554/eLife.31981.019
Author response https://doi.org/10.7554/eLife.31981.020

## Additional files

**Supplementary files**

• Transparent reporting form
DOI: https://doi.org/10.7554/eLife.31981.017

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
