## [Decision Letter]

Thank you for submitting your article "The yeast H^+^-ATPase Pma1 promotes Rag/Gtr-dependent TORC1 activation in response to H^+^-coupled nutrient uptake" for consideration by *eLife*. Your article has been reviewed by three peer reviewers, and the evaluation has been overseen by a Guest Reviewing Editor and Jonathan Cooper as the Senior Editor. The reviewers have opted to remain anonymous.

The reviewers have discussed the reviews with one another and the Reviewing Editor has drafted this decision to help you prepare a revised submission.

Summary:

The reviewers agree that the study is a well-designed set of technically sound experiments, with the exception of the quality of some Sch9 immunoblots (see comment 3). The manuscript provides an interesting and unusual explanation on how amino acid sufficiency is transmitted to TORC1 via Gtr1/2 in *S. cerevisiae*. We have decided to recommend the manuscript for publication in *eLife* after major revision. Specific comments are as follows.

1) The main conclusion of this work, i. e., proton influx coupled to nutrient uptake stimulates TORC1, would be significantly strengthened if the authors performed an experiment expressing (preferably in a *gap1∆* background) either of two non-yeast solute transporters that pump the same solute but one as a proton symporter and the other through a different mechanism (sodium symporter, ATP-dependent transporter, etc.) and investigate whether only the proton symporter activates TORC1. Inspiration for such an experimental setup might be taken from Nicklin P, et al. Cell. 2009; 36(3):521-34.

2) It is unclear which amino acids require Pma1 to activate TORC1 and which do not. This can be addressed experimentally. The authors should monitor TORC1 activity over time (with emphasis on the early time points) in ScPma1 and (Np)Pma4^822ochre^ expressing cells in response to different amino acids such as leucine (activates TORC1 via Gtr1/2) or glutamine (activates TORC1 also independently of Gtr1/2). They should ideally use *gtr1∆gtr2∆* cells as a control.

3) Figures 1E, 5C, 5G, Figure 7—figure supplement 1B and Figure 1—figure supplement 2. Total Sch9 levels are not controlled properly. Figure 1—figure supplement 2 shows that the antiserum against dephosphorylated Sch9 is sensitive to phosphorylation. Sch9 immunoblots should be repeated with better reagents.

4) The authors should determine whether Pib2 or SEACIT/SEACAT components are necessary for proton influx/Pma1 to activate TORC1. Gtr1/2 and Pib2 are believed to function independently, but the evidence does not rule out dependencies. Also, the SEACAT and SEACIT complexes regulate Gtr1/2 and TORC1. Single knockout mutants could be generated and analyzed.

5) The role of Pma1 in TORC1 activation is intriguing. The use of a separation of function allele of Pma1 that loses the ability to signal to TORC1 would be a good complement to this work. Perhaps a screen for viable Pma1 alleles that present an EGO phenotype (i.e., fail to recover from rapamycin growth arrest) could yield such an allele. This would not be a make-or-break issue for publication but a suggestion to improve the manuscript.

6) It is known that TORC1 is inactivated when cells are starved for a single amino acid such as leucine (in the case of *leu2∆* cells) or for phosphate, sulfate or zinc. When a single nutrient is lacking, other nutrients are abundant and, in principle, should be able to promote proton influx. The authors should comment about this in the Discussion.

7) Figures 4G and 4H. FCCP causes cytosolic acidification in 2 min and pH remains almost constant for 30 min. However, Npr1 phosphorylation increases gradually during 30 min of FCCP treatment. How can the authors explain this difference?

8) Figures 4I and 6K. Npr1 phosphorylation is largely, but not completely, impaired in the *gtr1∆ gtr2∆* strain. The authors should provide an explanation for this residual phosphorylation. This could be discussed.

9) Subsection “Inhibition of the vacuolar V-ATPase activates TORC1”. Optional: Since the authors state that glucose-fermenting cells have a high capacity to buffer the cytosol, how does TORC1 activity respond in respiring cells? This could be discussed.

10) The authors may wish to discuss their results in light of the recent work demonstrating TORC1 inactivation mediated via assembly into a TOROID (Prouteau M, et al., 2017).

11) Supplementary data is not always appropriately cited from the main text.

12) Introduction, second paragraph. "RhEB" should be Rheb.

13) Introduction, third paragraph. TOR was originally identified in yeast in 1991. TORC1 was also identified in yeast, but in 2002 (Loewith R, et al. Mol Cell. 2002; 10(3):457-68).

14) Introduction, third paragraph. In the genetic nomenclature of *S. cerevisiae*, a dominant allele is denoted by using uppercase italics. Thus, *tor1* should be TOR1.

15) Introduction, third paragraph. The authors incorrectly refer to TORC1 as "TORC1 complex". TORC1 is TOR Complex 1. Thus, it is redundant to write "TORC1 complex".

16) Introduction, third paragraph. The kinase that phosphorylates Rps6 in vivo is not Sch9, but Ypk3 (González A, PLoS One. 2015;10(3):e0120250 and Yerlikaya S, et al. Mol Biol Cell. 2016; 27(2):397-409).

17) Subsection “Uptake of ß-alanine via the Gap1 permease causes Rag/Gtr-dependent TORC1 activation without increasing internal pools of amino acids”. "CH_3_" should be corrected to "CH_2_" or "methylene group".

---

## [Author Response]

Summary:The reviewers agree that the study is a well-designed set of technically sound experiments, with the exception of the quality of some Sch9 immunoblots (see comment 3). The manuscript provides an interesting and unusual explanation on how amino acid sufficiency is transmitted to TORC1 via Gtr1/2 in S. cerevisiae. We have decided to recommend the manuscript for publication in eLife after major revision. Specific comments are as follows.1) The main conclusion of this work, i. e., proton influx coupled to nutrient uptake stimulates TORC1, would be significantly strengthened if the authors performed an experiment expressing (preferably in a gap1∆ background) either of two non-yeast solute transporters that pump the same solute but one as a proton symporter and the other through a different mechanism (sodium symporter, ATP-dependent transporter, etc.) and investigate whether only the proton symporter activates TORC1. Inspiration for such an experimental setup might be taken from Nicklin P, et al. Cell. 2009; 36(3):521-34.

We agree that this is an ideal experiment to strengthen the conclusion that it is the influx of H^+^ that triggers TORC1 activation. This experiment, however, is not easy as transporters from other species are most often poorly active when expressed in yeast. For instance, our first idea was to express in a *gap1 ssy1* mutant, defective in uptake of several amino acids including tryptophan, the human MCT10/TAT1 aromatic amino acid transporter. The group of Fumiyoshi Abe indeed reported that MCT10 expressed in yeast is functional and promotes transport of tryptophan via facilitated diffusion, without the need of an H^+^ gradient across the plasma membrane (Uemera et al. BBA vol. 1859, pp. 2076-2085). Dr. Abe kindly sent us a plasmid for expression of MCT10/TAT1 in yeast and we introduced it into the *gap1 ssy1* mutant. However, MCT10/TAT1 was unable to restore growth on aromatic amino acids as sole nitrogen source, indicating that it is not functional (as growth tests are indeed highly sensitive to detect amino-acid permease activities).

We thus reasoned that it is preferable to compare cells *endogenously* expressing a facilitator or an H^+^-coupled transporter, both recognizing the same compound. Hexoses including fructose are known to be transported into yeast cells via facilitators, namely the Hxt proteins. We contacted our colleague Eckhard Boles who drew our attention on the fact that particular strains of *S. cerevisiae* express an H^+^-coupled fructose transporter termed Fsy1. Dr. Paula Gonçalves (Lisbon) who reported this interesting finding kindly sent us strains and plasmids to express Fsy1 in the *hxt* null strain (EBY.VW4000) lacking the *HXT1 to -17* and *GAL2* genes, and thus unable to use glucose or fructose as a carbon source. We thus compared cells of the *hxt* null mutant expressing or not Fsy1. As a control, we used the wild-type CEN.PK2-1c strain (from which EBY.VW4000 derives) expressing the endogenous Hxt facilitators. The strains were first grown on maltose (they can use this carbon source as they still possess their maltose permease genes) and then shifted for a few hours on ethanol, because the *FSY1* gene is more highly expressed on ethanol (Rodrigues de Sousa et al., 2010). As in all other experiments, the nitrogen source was proline. Using these growth conditions, we then set up conditions of equivalent ^14^C-fructose uptake in Hxt- and Fsy1-expressing cells. As expected, no significant uptake was detected in the *hxt* null strain. Furthermore, we could nicely confirm that fructose uptake via Fsy1 is inhibited if cells are pretreated with FCCP, this being not the case when this uptake is mediated by the Hxt facilitators. We finally applied these growth and fructose uptake conditions to assay TORC1 activation using Sch9 phosphorylation as a readout. We observed a rapamycin-sensitive increased phosphorylation of Sch9, upon fructose uptake, in Fsy1-expressing cells. Such an activation of TORC1 was not detected in the wild-type incorporating fructose via the Hxt facilitators neither the *hxt* null strain expressing none fructose transporter. These results have now been included as Figures 4G, H, Iandare described in the last paragraph of the subsection “H^+^ influx coupled to transport promotes Rag/Gtr-dependent stimulation of TORC1”.

One comment must be added. The result of this experiment could seem surprising as previous studies reported that addition of glucose (entering cells via Hxt facilitators) to glucose-starved cells reactivates TORC1, assayed also using Sch9 phosphorylation as a readout. These experimental conditions, however, are different from the ones we have applied as we grew cells on maltose and transferred them to ethanol, two carbon sources cells can utilize. In these cells, Pma1 remained active as we could detect an FCCP-sensitive fructose uptake via Fsy1. This markedly contrasts with glucose-starved cells where Pma1 is inhibited and the H^+^ gradient at the plasma membrane collapses (see i. e., Figure 4A, 4B). Furthermore, when glucose is provided to glucose-starved cells, Pma1 is rapidly reactivated and this should promote H^+^-coupled uptake of nutrients including amino acids. This H^+^-coupled uptake could contribute to the glucose-elicited reactivation of TORC1. A comment emphasizing this point has also been included in Discussion (second paragraph).

We have tried to also use HA-Npr1 phosphorylation as a readout of TORC1 activation in response to fructose uptake. We were faced to unexpected difficulties that seem to originate from the fact that under the particular carbon supply conditions used in the experiment, Npr1 undergoes a high basal of phosphorylation largely resistant to rapamycin (data not shown). We have not further investigated this effect (because of a lack of time) and decided to focus on Sch9.

We thank the reviewers for their excellent suggestion as we think that this experiment strongly strengthens the main conclusion of our work, namely that H^+^ influx coupled to nutrient uptake is a signal for transient TORC1 activation in yeast.

2) It is unclear which amino acids require Pma1 to activate TORC1 and which do not. This can be addressed experimentally. The authors should monitor TORC1 activity over time (with emphasis on the early time points) in ScPma1 and (Np)Pma4^822ochre^ expressing cells in response to different amino acids such as leucine (activates TORC1 via Gtr1/2) or glutamine (activates TORC1 also independently of Gtr1/2). They should ideally use gtr1∆gtr2∆ cells as a control.

We set up conditions of equivalent initial uptake of Gln, Leu and Arg in the *(Sc)PMA1*- and *(Np)PMA4^822ochre^*-expressing cells (rem: for Arg, uptake was equivalent at least during the first 4 minutes). We then applied these conditions to analyze TORC1 activation using HA-Npr1 phosphorylation as a readout. The result showed that uptake of Gln, Leu or Arg stimulates TORC1 in (Sc)Pma1 cells (unless rapamycin was present), but not in the (Np)Pma4^822ochre^ cells. We additionally analyzed the influence of the *gtr1Δ gtr2Δ* mutations, as requested. None significant increased phosphorylation of HA-Npr1 phosphorylation was detected upon addition of Leu, Gln or Arg to the *gtr1Δ gtr2Δ* strain. These results thus support our conclusion that Pma1 plays an important role in Rag/Gtr-dependent TORC1 activation in response to amino-acid uptake. They have been included as Figure 7—figure supplement 2 in the revised version of our article (Figure 7 is former Figure 6). We describe them in the second paragraph of the subsection “TORC1 activation in response to increased cytosolic H^+^ requires the Pma1 H^+^ ATPase”.

3) Figures 1E, 5C, 5G, Figure 7—figure supplement 1B and Figure 1—figure supplement 2. Total Sch9 levels are not controlled properly. Figure 1—figure supplement 2 shows that the antiserum against dephosphorylated Sch9 is sensitive to phosphorylation. Sch9 immunoblots should be repeated with better reagents.

All indicated immunoblots have been either hybridized again, or the full experiment repeated, using an antibody against Sch9 (kindly provided by Dr. R. Loewith) that is not sensitive to phosphorylation. Furthermore, we decided to repeat the experiments for Figures 3C, 4D, 4J (now Figure 5C), 5C (now Figure 6C) and 5G (now Figure 6G) using the better-quality antibody. The latter was also used for Figure 6G (now Figure 7G). All obtained data fitted with our expectation and now appear in the revised version of our article.

4) The authors should determine whether Pib2 or SEACIT/SEACAT components are necessary for proton influx/Pma1 to activate TORC1. Gtr1/2 and Pib2 are believed to function independently, but the evidence does not rule out dependencies. Also, the SEACAT and SEACIT complexes regulate Gtr1/2 and TORC1. Single knockout mutants could be generated and analyzed.

We introduced in the strain background used in our study (Σ1278b) a deletion of the *PIB2, IML1* (SEACIT), and *SEH1* (SEACAT)genes. We then analyzed H^+^-influx-elicited TORC1 activation using FCCP as a trigger and HA-Npr1 phosphorylation as a readout. The results showed that:

- Pib2 is not essential to H^+^-influx-elicited TORC1 stimulation.

- In the *ilm1Δ* mutant, HA-Npr1 displayed a high basal phosphorylation, as expected. Addition of FCCP further increased HA-Npr1 phosphorylation but only after a long incubation time (30 min). This phenotype is similar to the one observed in the *gtr1Δ gtr2Δ* mutant (see new Figure 5D) and show that FCCP also promotes a late TORC1 activation in a manner independent on the Gtr1/2 GTPases and their upstream regulators. This particular point, namely the existence of a late, Gtr-independent, activation of TORC1 upon FCCP addition, has also been raised in comments 7 and 8 (see below our responses).

- In the *seh1Δ* mutant, FCCP-elicited activation of TORC1 was impaired, at least during the early minutes after FCCP addition. We also noticed that the intensity of the HA-Npr1 signal is much weaker in this strain. This effect was also observed (though less pronounced) in the *gtr1Δ gtr2Δ* (see i.e., Figures 1G, 4E, and new 5D) and *tco89Δ* (our unpublished data) mutants, but its cause remains unclear. It in fact suggests that HA-Npr1 is less abundant when TORC1 is poorly active. Again, addition of FCCP to this strain further increased HA-Npr1 phosphorylation but only after a long incubation time (30 min), as in the *gtr1Δ gtr2Δ* mutant (see comments 7 and 8 below).

In conclusion, Pib2 is not involved in TORC1 activation in response to H^+^-influx. In contrast, components of the SEACAT and SEACIT complexes seem to play an important role in this activation. Overall these results support the proposed view that TORC1 activation in response to H^+^ influx requires the Rag/Gtr GTPases and their SEAC upstream regulators. Pib2, however, already known to act independently of the Gtr1/2, is not essential to this TORC1 activation pathway. These additional results now appear as Figures 5E, 5F, and 5G in the revised version of our article. They are commented on in the second paragraph of the subsection “H^+^ diffusion via a protonophore promotes Rag/Gtr-dependent stimulation of TORC1”.

5) The role of Pma1 in TORC1 activation is intriguing. The use of a separation of function allele of Pma1 that loses the ability to signal to TORC1 would be a good complement to this work. Perhaps a screen for viable Pma1 alleles that present an EGO phenotype (i.e., fail to recover from rapamycin growth arrest) could yield such an allele. This would not be a make-or-break issue for publication but a suggestion to improve the manuscript.

We agree that such allele would beautifully complement our data suggesting a potential regulatory function of Pma1 in TORC1 activation in response to H^+^ influx. We thus generated a library of random *pma1* mutants and, after transformation into the *GAL-PMA1 pma2* strain, we tried to positively select on a glucose medium colonies exhibiting increased resistance to rapamycin. We thus tried to isolate Pma1 variants hyperstimulating TORC1. Our first attempts failed: we've obtained resistant clones, but not more than after transformation with a non-mutagenized *PMA1* gene. However, we are not 100% sure that our positive selection method was suited (quality of the library, selection conditions, …). We decided to temporarily put on hold this experiment (because of lack of time) and decided to invest in another approach.

We indeed found in the literature a very interesting article series showing that the Na^+^/K^+^-ATPase of animals cells, similar in sequence to the yeast Pma1 (both belong to the same P-type ATPase structural family), has signaling capabilities. Namely, the Na^+^/K^+^-ATPase is an inhibitor of the Src kinase, and binding of cardiotonic steroids such as ouabain to the Na^+^/K^+^-ATPase relieves this inhibition. Remarkably, the group of Zijian XIE identified a cytosolic region in the Na^+^/K^+^-ATPase which when mutated provoked a loss of interaction with Src, without affecting the catalytic activity of the ATPase (Lai et al., 2013). We found very interesting the observation that when the sequences of Pma1 and the Na^+^/K^+^-ATPase were aligned, this particular region appeared well conserved in Pma1. We thus generated several Pma1 mutants harboring substitutions in the depicted sequence, expressed them in the *GAL-pma1 pma2* mutant, and tested their functionality. These substitutions were: A408P, A411P, A411P+A604V, R414A, and C409A+L410A+R414A. Unfortunately, this approach did not lead to any relevant result: Pma1(A408P) turned out to be nonfunctional, and we did not observe any TORC1 activation defect in response to FCCP in cells expressing the other Pma1 mutants (data not shown).

We hope that the reviewers will appreciate that we did our best to also address this point. For the revised version of our article, we propose (i) to include a comment in Discussion about the illustrated role of the Na^+^/K^+^-ATPase in signaling (ii) to mention that we have tried to introduce into Pma1 substitutions in a similar region, but that these did not impair TORC1 activation (lines 592-602). We think that this comment will emphasize that although it seems reasonable to propose a model whereby Pma1 signals to TORC1 (given the properties of the Na^+^/K^+^-ATPase), it is far from being demonstrated and requires further investigation.

6) It is known that TORC1 is inactivated when cells are starved for a single amino acid such as leucine (in the case of leu2∆ cells) or for phosphate, sulfate or zinc. When a single nutrient is lacking, other nutrients are abundant and, in principle, should be able to promote proton influx. The authors should comment about this in the Discussion.

We think that two aspects must be considered to explain this apparent paradox. First, reduction of TORC1 activity in cells starved for a single nutrient is probably due to control mechanisms capable of sensing the nutritional status of the cell, and not just to a potential drop of its H^+^-coupled uptake. Consistently, addition to these cells of the limiting nutrient leads to its H^+^-coupled uptake associated with a transient reactivation of TORC1, but more sustained TORC1 activity requires proper assimilation of the nutrient by the cells. Second, it is likely that the drop of TORC1 activity caused by a single starvation not only promotes derepression of high-affinity permeases for the limiting nutrient, but also accelerated endocytosis of the permeases for the non-limiting nutrients. It is indeed known that inhibition of TORC1 by rapamycin promotes endocytosis of many different permeases. This general control is likely associated with a global drop of H^+^ influx that could also contribute to the reduction of TORC1 activity despite the abundance of other nutrients in the medium. These two aspects have now been better emphasized in the Discussion of the revised version of our article (second paragraph).

7) Figures 4G and 4H. FCCP causes cytosolic acidification in 2 min and pH remains almost constant for 30 min. However, Npr1 phosphorylation increases gradually during 30 min of FCCP treatment. How can the authors explain this difference?

It is true that HA-Npr1 phosphorylation further increases after a long incubation in the presence of FCCP (Figure 4H, now Figure 5B). This apparent late activation of TORC1 was also detected in the *gtr1Δ gtr2Δ* strain and in mutants lacking SEACIT or SEACAT upstream regulators of Gtr1/2 (see comment 4 above). Our interpretation is that a longer treatment of cells with FCCP stimulates another TORC1 activation mechanism that is not dependent on the Gtr1/2 GTPases neither their upstream regulators. This is reminiscent of the previous observation by M. Hall and collaborators (Stracka et al., 2014). One possibility is that this late activation of TORC1 is due to FCCP-induced release of amino acids or other compounds stored in the vacuole or mitochondria. This tentative interpretation has now been included in the revised version of the article (subsection “H^+^ diffusion via a protonophore promotes Rag/Gtr-dependent stimulation of TORC1”, second paragraph).

8) Figures 4I and 6K. Npr1 phosphorylation is largely, but not completely, impaired in the gtr1∆ gtr2∆ strain. The authors should provide an explanation for this residual phosphorylation. This could be discussed.

Figure 4I (now Figure 5D): as discussed above, we now propose in the text (subsection “H^+^ diffusion via a protonophore promotes Rag/Gtr-dependent stimulation of TORC1”, second paragraph) that this Gtr1/2-independent activation of TORC1 might be promoted by amino acids released from the vacuole or mitochondria.

Figure 6K (now Figure 7K): that phosphorylation of HA-Npr1 in response to NH_4_^+^ is not impaired in the *gtr1Δ gtr2Δ* strain is expected and consistent with the previous work of Stracka et al. (2014). We think that this was clearly mentioned in the text and does not need to be further emphasized. The interesting result of this experiment is that Pma1 does not play an important role in a TORC1 activation pathway known to be independent on the Gtr1/2 GTPases.

9) Subsection “Inhibition of the vacuolar V-ATPase activates TORC1”. Optional: Since the authors state that glucose-fermenting cells have a high capacity to buffer the cytosol, how does TORC1 activity respond in respiring cells? This could be discussed.

We could at least show that in cells shifted for six hours on ethanol, TORC1 is activated in response to H^+^-coupled fructose uptake but not passive fructose transport (see point 1). This observation suggests that the activity of TORC1 is also stimulated by H^+^ influx in respiring cells, but this question probably needs further investigation beyond the scope of our article.

10) The authors may wish to discuss their results in light of the recent work demonstrating TORC1 inactivation mediated via assembly into a TOROID (Prouteau M, et al., 2017).

This reference and a comment have now been included in the text (Discussion, sixth paragraph).

11) Supplementary data is not always appropriately cited from the main text.

This has now been corrected.

12) Introduction, second paragraph. "RhEB" should be Rheb.

This has now been corrected.

13) Introduction, third paragraph. TOR was originally identified in yeast in 1991. TORC1 was also identified in yeast, but in 2002 (Loewith R, et al. Mol Cell. 2002; 10(3):457-68).

This has now been corrected.

14) Introduction, third paragraph. In the genetic nomenclature of S. cerevisiae, a dominant allele is denoted by using uppercase italics. Thus, tor1 should be TOR1.

This has now been corrected.

15) Introduction, third paragraph. The authors incorrectly refer to TORC1 as "TORC1 complex". TORC1 is TOR Complex 1. Thus, it is redundant to write "TORC1 complex".

This has now been corrected.

*16) Introduction, third paragraph. The kinase that phosphorylates Rps6* in vivo *is not* Sch9*, but Ypk3 (González A, PLoS One. 2015;10(*3*):e0120250 and Yerlikaya S, et al. Mol Biol Cell. 2016; 27(*2*):397-409).*

This has now been corrected.

17) Subsection “Uptake of ß-alanine via the Gap1 permease causes Rag/Gtr-dependent TORC1 activation without increasing internal pools of amino acids”. "CH_3_" should be corrected to "CH_2_" or "methylene group".

This has now been corrected.